# Telomeric DNA damage response mediates neurotoxicity of Aβ42 oligomers in Alzheimer's disease

Sara Sepe [ID][1], Federica Rey [ID][1], Alexandra Mancheno-Ferris [ID][1], Alessandra Bigi [ID][2], Giulia Fani[2], Devid Damiani[3], Matteo Cabrini[1,4], Eugenia Marinelli[1], Julio Aguado[1,5], Liliana Contu[3], Alessia di Lillo[1], Sara Boggio[1], Sara Tavella[1,4], Ilaria Rosso[1], Stefano Gustincich[3], Fabrizio Chiti [ID][2] & Fabrizio d'Adda di Fagagna [ID][1,4][✉]

## Abstract

**Ageing is the major risk factor for Alzheimer's disease (AD), the most common neurodegenerative disorder. DNA damage is a hallmark of ageing, particularly when occurring at telomeres, genomic regions vulnerable to oxidative damage and often challenging for the cell to repair. Here, we show that brains of 3xTg-AD mice, an established AD model characterized by amyloid-β (Aβ)-induced pathology, exhibit increased activation of DNA damage response (DDR) pathways at telomeres. Exposure of mouse primary hippocampal neurons to 42-residue Aβ (Aβ42) oligomers, a significant pathogenetic contributor to AD, triggers telomeric DDR by increasing the levels of reactive oxygen species caused by calcium imbalance. Antisense oligonucleotides targeting non-coding RNAs generated at damaged telomeres in vivo (in 3xTg-AD mice) and in vitro reduce neurotoxicity in iPSC-derived human cortical neurons and mouse primary neurons while inhibiting Aβ42-induced telomeric DDR, and restore transcriptional pathways altered by Aβ and found dysregulated in AD patients. These results unveil an unexpected role of telomeric DNA damage responses in Alzheimer's disease pathogenesis, and suggest a novel target for the development of RNA-based therapies.**

**Keywords** Alzheimer's disease (AD); Aging; Telomeres; DNA damage response (DDR); ASO
**Subject Categories** DNA Replication, Recombination & Repair; Molecular Biology of Disease; Neuroscience

## Introduction

Alzheimer's disease (AD) is the most common form of dementia in adults over age 65 (2015 Alzheimer's disease facts and figures, 2015). It is a neurodegenerative disorder characterized by progressive memory loss and cognitive decline. Clinical symptoms typically include impairment of recent memory, language disturbances, and alterations in abstract reasoning, concentration and thought sequencing (executive function) (Masters et al, 2015). From a pathological point of view, AD is characterized by amyloid-β protein (Aβ) accumulation and plaque formation, intraneuronal neurofibrillary tangles composed by the hyperphosphorylated and polyubiquitinated tau protein (Knopman et al, 2021). Ageing is the primary risk factor for AD, and accumulating evidence implicates DNA damage and the activation of the DNA damage response (DDR) pathways as a key contributor to ageing and neurodegeneration (Schumacher et al, 2021; Hou et al, 2019). Protracted DDR activation, caused by accumulation of persistent DNA damage and/ or impaired DNA repair, can become pathologic and contribute to AD and age-related diseases (Thadathil et al, 2021; Welch and Tsai, 2022; Wilson et al, 2023; Herdy et al, 2022). Indeed, increased levels of DNA damage and DDR markers have been described in post-mortem brain tissues from AD patients and in AD murine models (Ranganathan et al, 2023; Simpson et al, 2015; Sykora et al, 2015; Chow and Herrup, 2015). The observation that patients, or mouse models, carrying genetic mutations in DNA repair genes often show prominent neurological phenotypes support the notion that DNA damage accumulation can trigger pathological changes in the central nervous system (Madabhushi et al, 2014).

DDR pathways are engaged upon DNA damage in physiological and pathological conditions. DDR factors accumulate at sites of DNA damage, forming focal structures nucleated by the phosphorylated form of histone H2AX (named γH2AX) accumulating, among others, the activated form of the protein kinase ataxia telangiectasia mutated (pATM) and its substrates, and 53BP1

[1]IFOM ETS - The AIRC Institute of Molecular Oncology, Milan, Italy. [2]Department of Experimental and Clinical Biomedical Sciences, Section of Biochemistry, University of Florence, Florence, Italy. [3]Center for Human Technologies, Non-coding RNAs and RNA-based Therapeutics, Istituto Italiano di Tecnologia (IIT), Genova, Italy. [4]Institute of Molecular Genetics (IGM), National Research Institute (CNR), Pavia, Italy. [5]Present address: University of Colorado Anschutz Medical Campus, Aurora, CO, USA.
[✉]E-mail: fabrizio.dadda@igm.cnr.it

(D'Adda Di Fagagna, 2008). DDR foci promote the signaling to downstream effectors such as the cyclin-dependent kinase (CDK) inhibitor p21 (D'Adda Di Fagagna, 2008), product of the CDKN1A gene. DNA damage is not equally repairable, and we and others have reported that DNA lesions occurring within telomeric repeats at chromosome ends resist endogenous repair activities and thus DNA damage persists (Fumagalli et al, 2012; Hewitt et al, 2012). Unrepaired persistent DNA damage causes prolonged DDR signaling that can lead to the establishment of cellular senescence (D'Adda Di Fagagna, 2008). Recently, RNA emerged as a key and direct contributor to DDR activation and foci formation (Francia et al, 2012; Michelini et al, 2017; D'Alessandro et al, 2018; Pessina et al, 2019). We and others have shown that DNA double-strand breaks (DSBs) recruit RNA polymerase II, resulting in the bidirectional transcription from exposed DNA ends of damage induced non-coding RNAs (Michelini et al, 2017; Pessina et al, 2019; Sharma et al, 2021). These RNAs are essential for full DDR activation and DDR foci assembly. Similarly, telomeric damage leads to the transcription of telomeric DNA sequences and thus accumulation of telomeric damage-induced non-coding RNA (tdincRNA) (Rossiello et al, 2017). Inhibition of tdincRNAs by sequence-specific telomeric antisense oligonucleotides (tASOs) impedes full DDR activation preventing DDR foci formation selectively at telomeres, as demonstrated in cultured cells and in vivo in mice (Rossiello et al, 2017; Aguado et al, 2019; Sepe et al, 2022).

During ageing and in age-related diseases telomeres have been reported to accumulate DNA damage and DDR markers in several tissues, including brain (Fumagalli et al, 2012; Hewitt et al, 2012). Consistent with this, a correlation between cognitive decline or dementia and telomere shortening in AD patients has been reported (Cao et al, 2023). In mouse models, critically short telomeres, which directly trigger telomeric DDR (tDDR) activation, have been shown to worsen AD pathology (Suelves et al, 2023), and loss of p21, a CDK inhibitor induced by DNA damage, can rescue cognitive deficits and neuronal dysfunction in mice with shortened telomeres (Jurk et al, 2012).

Oxidative stress is a known major feature of AD pathology. Indeed, individuals with established AD-related cognitive dysfunction were shown to have an imbalance in oxidant/antioxidant systems, with an increase in radical oxygen species (ROS) levels (Massaad and Klann, 2011; Fanelli et al, 2013; Porcellotti et al, 2015). Telomeric DNA is particularly susceptible to oxidative damage due to its guanine-rich composition and it is a preferential site of 8-oxoG accumulation (Von Zglinicki, 2002; Henle et al, 1999; Oikawa et al, 2001; Steenken and Jovanovic, 1997). These lesions can be converted into DSBs and accumulate over a lifetime (Polyzos et al, 2024) that may persist due to their poor repairability (Fumagalli et al, 2012; Hewitt et al, 2012). The gradual accumulation of telomeric DNA damage and sustained tDDR signaling is therefore considered a significant contributor to age-related pathologies (Fumagalli et al, 2012).

In AD, Aβ oligomers are the primary pathological determinant of AD and are considered the most toxic intermediates in Aβ plaque formation (Sakono and Zako, 2010; Kayed and Lasagna-Reeves, 2013; Sengupta et al, 2016; Balducci et al, 2010; Martins et al, 2008; Koffie et al, 2009). One of the mechanisms through which they have been shown to cause toxicity is by eliciting an abnormal and *N*-methyl-D-aspartate (NMDA) activation that

disrupts neuronal calcium ($Ca^{2+}$) homeostasis, increasing oxidative stress and mitochondrial dysfunction. Indeed, increased ROS levels due to impaired $Ca^{2+}$ homeostasis (Fani et al, 2022; De Felice et al, 2007) is likely due to the augmented activation of mitochondrial respiration necessary to restore physiological intracellular $Ca^{2+}$ levels. Therefore, Aβ42 oligomers have been associated with neuronal cell death and one mechanism proposed is mediated by Bax, a protein of the Bcl-2 gene family (Paradis et al, 1996; Su et al, 1997; Gao et al, 2016; Callens et al, 2021) also involved in DDR and mitochondrial dysfunction (Victorelli et al, 2023). Notably, Aβ42 oligomers have been shown to be sufficient to induce DDR markers in neuronal cultures, linking Aβ pathology directly to DNA damage (Suberbielle et al, 2013; Jung et al, 2016). In this study, we investigated the link between ageing and AD. Using an AD mouse model, as well as mouse primary hippocampal neurons and human cortical neurons derived from induced pluripotent stem cells (iPSCs), we unveiled a role for tDDR in mediating Aβ42 oligomer-induced neurotoxicity.

## Results

### Telomeric DNA damage and tdincRNA are increased in the brain of a mouse model of AD

The 3xTg-AD mouse model harbors three mutations associated with the familial form of AD (APP Swedish, MAPT P301L, and PSEN1 M146V) and it is an established animal model for AD which develops amyloid and tau-AD neuropathology, leading to memory and learning deficits (Billings et al, 2005). To determine the accumulation of DNA damage and the engagement of DDR pathways in this model, we analyzed cerebral cortex, namely isocortex and hippocampal region (specifically the Ammon's horn) from age-matched 12-month-old wild-type (wt) and 3xTg-AD mice by immunofluorescence with antibodies against γH2AX and 53BP1, two independent and robust markers of DDR activation combined with an antibody against MAP2 to label neurons. Software-based quantification of nuclear focal signals revealed an increased number of γH2AX and 53BP1 foci in both brain regions of 3xTg-AD mice compared with wt controls (Figs. 1A and EV1A) consistent with previous reports (Sykora et al, 2015). Since DNA damage at the telomeres tends to persist because of its intrinsic irreparability (Fumagalli et al, 2012; Hewitt et al, 2012), we tested whether DDR foci preferentially accumulated at telomeres. We therefore performed ImmunoFISH experiments on the brain combining DDR staining for γH2AX and telomeric DNA detection by a fluorescent peptide-nucleic acid (PNA) probe together with the staining for neurons using an antibody against MAP2; as control we localized DDR at centromeres, another repetitive DNA chromosomal region performing double staining using the antibody against γH2AX, a centromeric protein (CREST) and the neuronal marker MAP2. These analyses revealed that the DDR markers preferentially accumulate at telomeric regions in the isocortex of 3xTg-AD compared with wt (Fig. 1B,C), but not at centromeres (Fig. 1B,C). The analysis on the fraction of DDR positive telomeres or centromeres demonstrates that the telomeres endure increased damage in the isocortex of 3xTg-AD compared with wt (Fig. 1D).

We have previously reported that full DDR activation depends on RNA species synthesized at DSB (Francia et al, 2012; Michelini

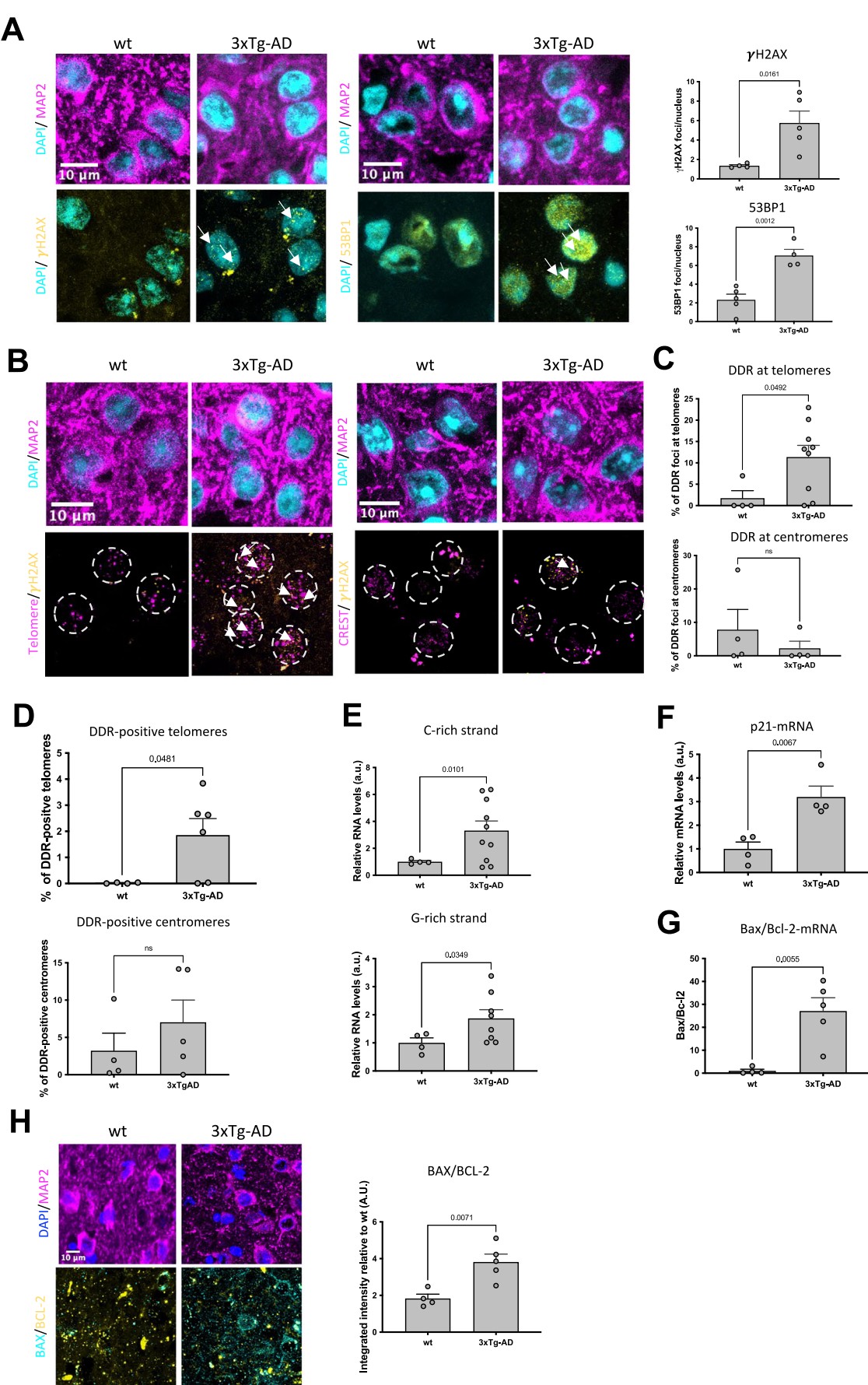

**Figure 1.  Markers of tDDR activation and pro-apoptotic gene expression are elevated in the brain cortex of 3xTg-AD mice.**

(A–H) Brain sections from 12 months old wt and 3xTg-AD mice were analyzed. Each circle in the graphs represents one mouse analyzed. (A) Representative confocal microscopy images of immunofluorescence staining of isocortex from wt and 3xTg-AD mice with antibodies against γH2AX, 53BP1, MAP2 and nuclei were stained with DAPI. Bar graph shows the quantification of the average number of γH2AX foci per nucleus ($n = 4$ wt and $n = 5$ 3xTg-AD), and 53BP1 foci per nucleus ($n = 5$ wt and $n = 4$ 3xTg-AD). Arrows point to γH2AX or 53BP1 foci. At least 50 nuclei per mouse section were scored. (B) Representative confocal microscopy images of ImmunoFISH staining combining γH2AX, MAP2 antibodies and FISH for telomeric DNA using a complementary PNA probe (left panel) and co-immunofluorescence with antibodies against γH2AX, MAP2 and the centromeric protein CREST (right panel) of sections from wt and 3xTg-AD mice. Nuclei were stained with DAPI and showed as dash circle in the lower panel. The arrows indicate the colocalizing events. At least 50 nuclei per mouse section were scored. (C) Bar graphs show the percentage of DDR foci (γH2AX foci signal) that localize at telomeres (telomeric PNA probe signal, upper graph)- or centromeres (CREST signal, lower graph)- as detected by immunoFISH ($n = 4$ for wt and $n = 9$ 3xTg-AD) or immunofluorescence ($n = 4$ for wt and $n = 4$ for 3xTg-AD mice), respectively. At least 50 nuclei per mouse section were scored. (D) Bar graphs show the percentage of telomeres—$n = 4$ for wt and $n = 6$ for 3xTg-AD—(telomeric PNA probe signal, upper graph) or centromeres—4 wt animals and 3xTg-AD mice—(CREST signal, lower graph)—colocalizing with DDR foci (γH2AX foci signal) as detected by immunoFISH or immunofluorescence, respectively. At least 50 nuclei per mouse section were scored. (E) Bar graphs show the quantification of G-rich and C-rich tdincRNA levels in 3xTg-AD ($n = 9/10$) mice relative to wt ($n = 4$) samples as measured by RT-qPCR. (F) Bar graphs show the quantification of p21 mRNA levels in 3XTg-AD ($n = 4$) mice relative to wt samples ($n = 4$) in isocortex as measured by RT-qPCR. (G) Bar graph represents the quantification of the ratio between the mRNA levels of Bax over Bcl-2 in 3xTg-AD ($n = 5$) mice relative to wt ($n = 4$) samples in isocortex as measured by RT-qPCR. (H) Representative confocal microscopy images of Immunofluorescence staining combining antibodies against MAP2, BAX and BCL-2 of sections from wt and 3xTg-AD mice (left panel). Bar graph represents the quantification of the ratio between the signal of Bax over BCL-2 from immunofluorescence staining in 3xTg-AD ($n = 5$) mice relative to wt ($n = 4$) samples in isocortex. For all graphs, unpaired $t$ test was applied. Data are represented as mean ± SEM. Source data are available online for this figure.

et al, 2017; D'Alessandro et al, 2018). Similarly, damaged telomeres trigger their transcription, generating tdincRNAs, which are essential for full telomeric DDR (tDDR) activation (Rossiello et al, 2017). The measurement by reverse transcription quantitative PCR (RT-qPCR) of tdincRNAs generated from both telomeric DNA strands (G-rich and C-rich strand) in both brain areas revealed higher levels in 3xTg-AD compared with wt mice (Figs. 1E and EV1B), independently strengthening the conclusions reached by imaging studies.

These observations prompted us to analyze in both region of the brain (Figs. 1F and EV1C) the levels of p21, a downstream effector in the DDR pathway. RT-qPCR analyses revealed a significant increase in p21 mRNA levels in the isocortex of 3xTg-AD relative to wt mice (Fig. 1F). In addition, we observed by RT-qPCR that in both isocortex and hippocampal region of this mouse model, the ratio between mRNA levels of the pro-apoptotic gene Bax over the anti-apoptotic gene Bcl-2 increase in the brain of the 3xTg-AD mice relative to wt mice (Figs. 1G and EV1D). Consistent with mRNA analyses results, immunofluorescence stainings revealed that the ratio between the protein level of BAX over BCL-2 was elevated in both brain regions analyzed. The signal was predominantly observed in neurons stained with the MAP2 marker (Figs. 1H and EV1E). These findings suggest that tDDR is associated with pro-apoptotic signaling in this context.

In summary, these results indicate that the areas of the brain most affected in AD exhibit a preferential accumulation of DDR at telomeres, that engages the downstream effector p21, at least in the isocortex, and the pro-apoptotic factor Bax.

## Aβ42 oligomers induce telomeric DDR in mouse primary neurons

Intraneuronal Aβ42 oligomers accumulation has been shown to be an important pathogenic mechanism for the onset of cognitive deficits in the 3xTg-AD mouse model preceding amyloid plaque formation (Billings et al, 2005). In order to determine whether Aβ42 oligomers are sufficient to cause neuronal telomeric DNA damage and to explore the mechanisms underlying this process, we treated mouse hippocampal primary neurons with Aβ42 oligomers,

the major toxic forms during the amyloid plaque formation process.

Aβ42 oligomers were prepared as described in Fig. EV2A and "Methods and protocols" section, following the method described in Beeg et al (2011). In all experiments described in this manuscript, Aβ42 oligomers were freshly prepared and each time tested before use by immunoblotting with anti-β (6E10) and anti-oligomer A11 antibodies which recognize the specific peptide sequence and its oligomeric structure, respectively, and Coomassie-based-stained polyacrylamide gel electrophoresis (Fig. EV2B).

We observed that the administration of Aβ42 oligomers in the culture medium of mouse hippocampal primary neurons is sufficient to trigger DDR activation, as demonstrated by the focal accumulation and activation of DDR factors such as γH2AX, 53BP1 and pATM (Fig. 2A), key components of the DDR cascade in mouse primary neurons as labeled with MAP2, a robust neuronal marker. This was specific for the amyloidogenic peptide Aβ42, since a control peptide did not cause any increase of DDR markers (Fig. EV2C).

We next investigated whether DDR preferentially accumulates at telomeres under these experimental conditions by conducting a colocalization study on mouse primary neurons. We performed ImmunoFISH to examine DDR markers at telomeres using a fluorescent PNA probe complementary to the telomeric sequence along with an antibody against γH2AX for DDR detection and against MAP2 to identify neurons. In parallel, we performed a double staining with antibodies against the centromeric protein (CREST) to study structurally similar regions as an internal control combined with antibodies against γH2AX and MAP2. We observed a distinct accumulation of γH2AX at telomeres in Aβ42 oligomers-treated neurons compared with vehicle-treated neurons; this accumulation was not observed when we analyzed centromeric markers (Fig. 2B,C). The analysis of the fraction of DDR-positive telomeres or centromeres shows that the telomeres of neurons treated with Aβ42 oligomers are preferentially damaged when compared with vehicle-treated neurons (Fig. 2D). Indeed, telomeres represent only about 0.1% of the total genome, thus any random co-localization should then be extremely rare. The observed accumulation of DDR markers at telomeres indicates that, relative

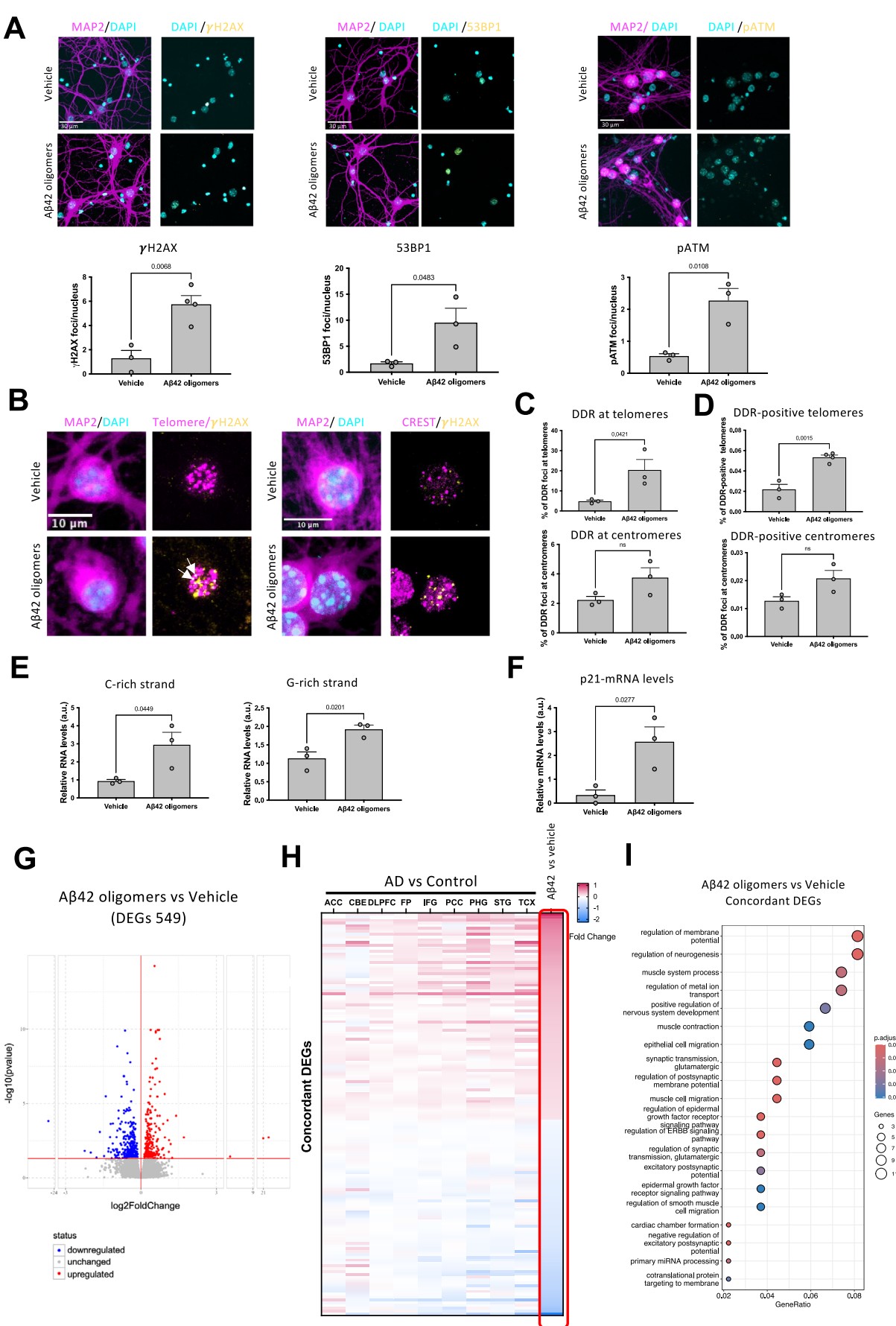

◄ **Figure 2.  Aβ42 oligomers induce DDR markers accumulation at telomeres in mouse hippocampal primary neurons.**

(A–I) Mouse hippocampal primary neurons were treated with vehicle or 10 μM Aβ42 oligomers for 48 h. Each circle represents an independent biological replicate. (A) Representative confocal image of immunofluorescence staining of primary neurons with antibodies against MAP2 (magenta) as neuronal marker, γH2AX, 53BP1 and pATM (yellow) as DDR markers, nuclei were stained with DAPI. Bar graph shows the quantification of the average number of γH2AX, 53BP1 and pATM foci per nucleus (for each condition $n = 3/4$). At least 50 cells for each condition in each replicate were analyzed. (B) Representative confocal microscopy images of immunoFISH combining γH2AX antibody and FISH for telomeric DNA using a complementary PNA probe (left panel) and double immunofluorescence combining antibodies against γH2AX and the centromeric protein CREST (right panel) of primary neurons. Arrows point to co-localization signals between γH2AX antibody and PNA probe. (C) Bar graphs show the percentage of DDR foci (γH2AX foci signal) that localize at telomeres (telomeric PNA probe signal - white arrows) or centromeres (CREST signal) as detected by immunoFISH or immunofluorescence, respectively ($n = 3$). At least 50 cells for each condition in each replicate were analyzed. (D) Bar graphs show the percentage of telomeres (telomeric PNA probe signal)- or centromeres (CREST signal) colocalizing with DDR foci (γH2AX foci signal) as detected by immunoFISH (Vehicle $n = 3$ and Aβ42 oligomers $n = 4$) or immunofluorescence (Vehicle $n = 3$ and Aβ42 oligomers $n = 3$), respectively. At least 50 cells for each condition in each replicate were analyzed. (E) Bar graphs show the quantification of G-rich and C-rich tdincRNA levels in Aβ42 oligomers treated primary neurons relative to vehicle treated primary neurons samples as measured by RT-qPCR. $n = 3$. (F) Bar graph shows the quantification by RT-qPCR of p21 mRNA in Aβ42 oligomers treated primary neurons relative to vehicle treated primary neurons. $n = 3$. (G) RNA-sequencing analysis was performed on $n = 2$ replicates for Aβ samples and $n = 2$ replicates for Vehicle conditions. Volcano plot showing the Differentially expressed genes (DEGs) in Aβ42 oligomer-treated vs vehicle-treated primary neurons. Each point represents a gene. The differential analysis was performed on R with DESeq2 (Love et al, 2014), DEGs were defined as genes with $p$ value adjusted (Benjamini–Hochberg) < 0.05 (genes with logFC >0 are considerate as upregulated, genes with logFC <0 as downregulated). (H) Heatmap showing the 142 DEGs in Aβ42 oligomers-treated vs vehicle-treated primary mouse hippocampal neurons that are concordant with human AD patients vs control patients in the indicated brain areas (concordant DEGs). ACC anterior cingulate cortex, CBE cerebellum, DLPFC dorsolateral prefrontal cortex, FP frontal pole, IFG inferior frontal gyrus, PCC posterior cingulate cortex, PHG parahippocampal gyrus, STG superior temporal gyrus, TCX temporal cortex. (I) The dotplot shows the 20 most significant Gene Ontology Biological processes that emerged from the enrichment analysis of "concordant DEGs" performed with ClusterProfiler. The name of the pathway is reported on the left of the graph, the $x$-axis represents the gene ratio, dots size represents the number of DEGs present in the pathway and the color indicates the adjusted $p$ value of the pathway enrichment. Statistical analysis was performed with over-representation analysis for Gene Ontology Biological processes on DEGs, performed with ClusterProfiler (Wu et al, 2021). Unpaired $t$ test was applied and data are represented as mean ± SEM in (A–F). Source data are available online for this figure.

to their abundance, telomeres are preferentially damaged by Aβ42 oligomers.

Similar results were observed when we treated HT-22 immortalized neuronal cell line with Aβ42 oligomers (Fig. EV3A–C). To further investigate the tDDR activation by the analysis of tdincRNA levels upon Aβ42 oligomers, we performed RT-qPCR and we detected increased levels of G-rich and C-rich strands of tdincRNAs in neurons treated with Aβ42 oligomers compared with controls (Fig. 2E). In addition, we observed an increased levels of p21 mRNA in mouse neurons treated with Aβ42 oligomers compared with vehicle (Fig. 2F), similar to results obtained in the isocortex of 3xTgAD.

In summary, we demonstrated that Aβ42 oligomers induce DDR activation in mouse hippocampal primary neurons, and that telomeres are preferential sites of DDR recognition as independently determined by colocalization studies and tdincRNA levels.

## Aβ42 oligomers induce gene expression deregulation in mouse primary neurons similar to those observed in human AD brains

To investigate the impact of Aβ42 oligomers treatment on gene expression, we performed next-generation RNA sequencing (RNA-seq) analysis of mouse primary neurons treated either with vehicle or with Aβ42 oligomers. The administration of Aβ42 oligomers to mouse primary neurons significantly deregulated 549 genes (indicated as differentially express gene (DEGs)) when compared with vehicle only-treated neurons (Fig. 2G). In addition, Gene Ontology (GO) Biological processes analysis revealed that several biological processes are impacted by Aβ42 oligomers (Fig. EV4A). Further inspection of GO biological process revealed that DNA repair pathways were not significantly dysregulated, suggesting that the abnormal DDR activation induced by Aβ42 oligomers is unlikely to be due to impaired DNA repair system.

In order to investigate whether Aβ42 oligomers treatment induces an AD-related signature, the set of 549 DEGs were subjected to "Gene Set Evaluation" function on AlzCode, a platform for multiview analysis of genes related to AD (Lin et al, 2022). We observed that Aβ42 oligomers treatment versus vehicle DEGs were significantly associated with AD profile gene expression in patients (Fig. EV4B,C). Through the Agora AD Knowledge Portal (Greenwood et al, 2020), we discovered that the Aβ42 oligomers versus vehicle DEGs were also dysregulated in specific brain areas from AD patients compared to controls (Fig. EV4D). Of these, 142 DEGs exhibited a shared dysregulation pattern in at least five human brain areas collected from post-mortem tissue of AD patients versus control—we named this set of DEGs as "concordant DEGs" (Fig. 2H). GO Biological processes analysis of these "concordant DEGs" revealed that they are involved in several relevant neuronal pathways (Fig. 2I).

In conclusion, transcriptomics analysis revealed that Aβ42 oligomers induce gene expression changes that are also observed in human AD, further supporting the use of this system to study AD pathogenesis.

## DDR activation at telomeres is mediated by ROS

Aβ42 oligomers have been reported to impact the neuronal redox state, leading to the induction of oxidative stress caused by the generation of ROS (De Felice et al, 2007; Fani et al, 2022). This prompted us to investigate if ROS could be the cause of the observed DDR induction triggered by Aβ42 oligomers. To test this, we included the antioxidant and radical scavenging agent *N*-acetyl cysteine (NAC) (Ezeriņa et al, 2018) in the culture medium prior to treatment with Aβ42 oligomers. The quantification of the signal generated by the CM-H₂DCFDA probe, an indicator of intracellular ROS in cells, showed that in this system Aβ42 oligomers increase ROS levels and that NAC reduces Aβ42 oligomers-induced ROS (Fig. 3A,B). Decrease ROS levels correlate with

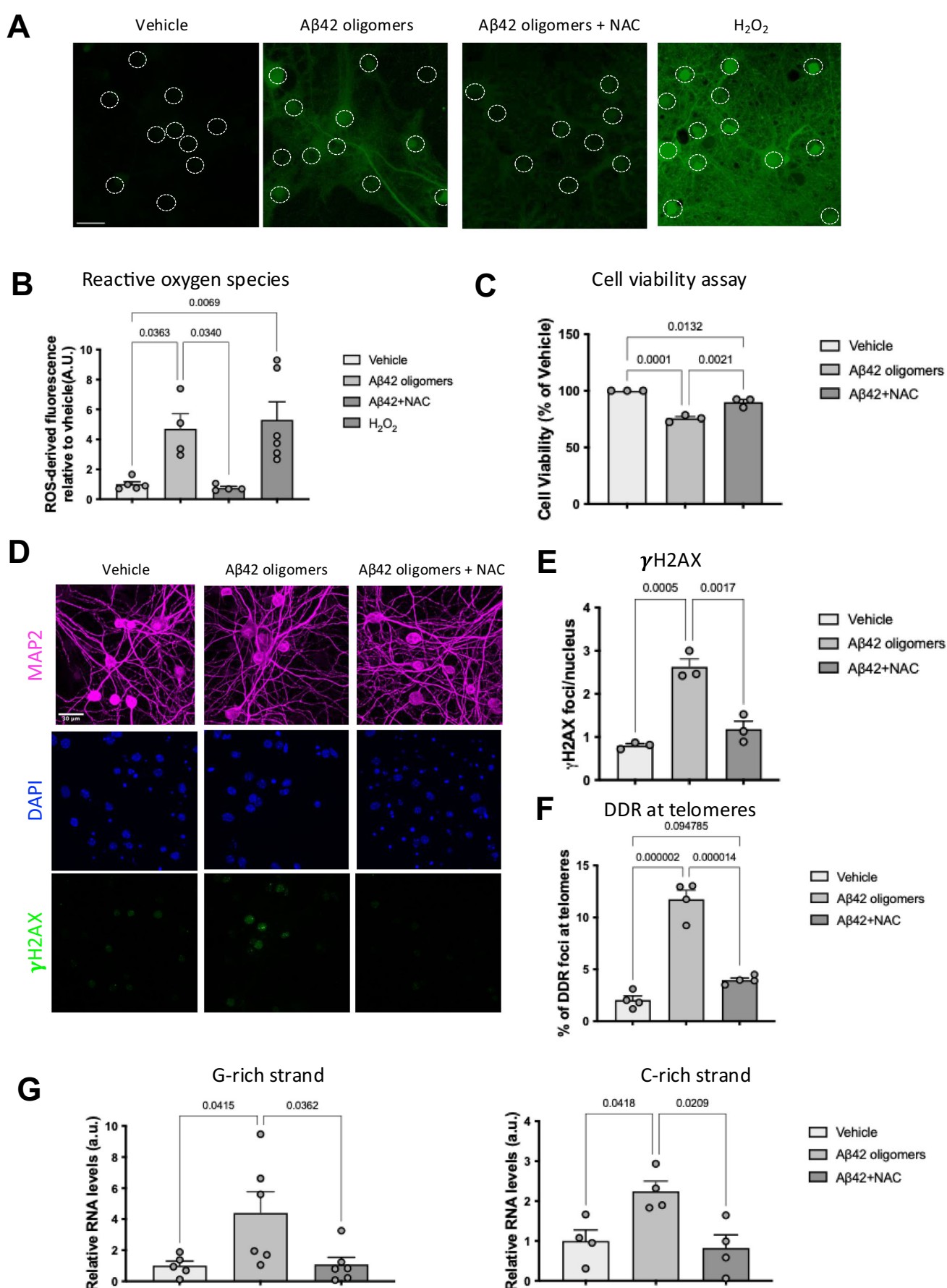

**Figure 3.** Reactive oxygen species mediate the Aβ42 oligomers-induced DDR at telomeres in mouse hippocampal primary neurons.

(A–G) Mouse primary neurons were treated for 30 min with 1 mM *N*-acetyl cysteine (NAC) before exposing them to Aβ42 oligomers or vehicle. Each circle represents an independent biological replicate. (A) Representative confocal microscopy images of primary neurons incubated with CM-H$_2$DCFDA probe, and the green fluorescence arises from the oxidation of the CM-H$_2$DCFDA probe. White dashed-line circles indicate the nuclei of neurons. Bar correspond to 20 μm. (B) Bar graph represents semiquantitative analysis of intracellular ROS detected by CM-H$_2$DCFDA. At least 40–50 cells for each condition in each replicate were analyzed (Vehicle, *n* = 5; Aβ42 oligomers, *n* = 4; Aβ42 + NAC, *n* = 4; H$_2$O$_2$, *n* = 6). (C) Bar graph represents cell viability as determined by CellTiter-Glo shown as a percentage of surviving cells compared to the vehicle. (Vehicle, *n* = 3; Aβ42 oligomers, *n* = 3; Aβ42 + NAC, *n* = 3). (D) Representative confocal images of immunofluorescence staining of primary neurons with antibodies against MAP2 (magenta) as neuronal marker, γH2AX (green) as DDR markers, nuclei were stained with DAPI. (E) Bar graph shows the quantification of average number of DDR foci (γH2AX foci signal) per nucleus. At least 50 cells for each condition in each replicate were analyzed. (Vehicle, *n* = 3; Aβ42 oligomers, *n* = 3; Aβ42 + NAC, *n* = 3). (F) Bar graphs show the percentage of DDR foci (γH2AX foci signal) that localize at telomeres (telomeric PNA probe signal) as detected by immunoFISH. (Vehicle, *n* = 4; Aβ42 oligomers, *n* = 4; Aβ42 + NAC, *n* = 4). (G) Bar graphs show the quantification of G-rich and C-rich tdincRNA levels relative to vehicle-treated mouse primary neurons samples as measured by RT-qPCR. (Vehicle, *n* = 5; Aβ42 oligomers, *n* = 6; Aβ42 + NAC, *n* = 6 for G-rich strand and Vehicle, *n* = 4; Aβ42 oligomers, *n* = 4; Aβ42 + NAC, *n* = 4 for C-rich strand. Ordinary one-way ANOVA was applied, and data are represented as mean ± SEM. Source data are available online for this figure.

improvement in neurons viability following NAC treatment (Fig. 3C), as assessed by CellTiter-Glo luminescent cell viability assay based on ATP detection.

To test the role of ROS in DDR activation, we studied DDR activation in this experimental setting. We observed that NAC-mediated ROS reduction leads to a decrease in the average number of total DDR foci per nucleus (Fig. 3C,D), and an apparently increased reduction of tDDR, as determined by colocalization study (Fig. 3E). Consistently, we observed that the levels of tdincRNAs are reduced by NAC treatment (Fig. 3F).

These results demonstrate that DDR, including tDDR activation, is caused by ROS generated upon exposure of mouse primary neurons to Aβ42 oligomers.

## ROS-induced DDR activation upon Aβ42 oligomers is dependent on altered Ca²⁺ flux in neurons

We next sought to determine the mechanisms by which Aβ42 oligomers increase ROS in mouse hippocampal primary neurons. It has been reported that Aβ42 oligomers target and activate the NMDA/AMPA receptors, leading to a rise in intracellular Ca²⁺ levels (Koffie et al, 2009; Fani et al, 2022; De Felice et al, 2007). In this context, memantine (an NMDA-receptor channel blocker) can counteract some of the effects of Aβ42oligomers (De Felice et al, 2007) and for this reason it is an Food and Drug Agency (FDA)-approved treatment for AD symptoms in patients (del Río-Sancho, 2020). When mouse hippocampal primary neurons were treated with Aβ42 oligomers, we detected an increase of intracellular Ca²⁺ as measured by fluorescence FLUO-4 AM probes (Fig. 4A)—ionomycin was used as positive control (Fig. EV5A,B). The observed rise in Ca²⁺ levels was the result of Ca²⁺ influx mediated by NMDA and AMPA receptors, since NMDA-receptor channel blocker memantine and AMPA-receptor channel blocker CNQX prevented it (Fig. 4A). In order to determine the source of Ca²⁺, which can commonly be stored extracellularly, we included the Ca²⁺ chelator EGTA in the cell culture medium: we observed it prevented Aβ42 oligomers-induced increase of intracellular Ca²⁺ indicating that the source of Ca²⁺ is extracellular (Fig. 4A).

In neurons, increase in intracellular Ca²⁺ levels are often accompanied by the activation of Ca²⁺-expelling pumps. This process is energy-demanding on mitochondria (Bianchi et al, 2004). In addition, an NMDA/AMPA receptors induced rise of Ca²⁺ levels activates NADPH oxidase with further ROS formation (Brennan

et al, 2009). To determine whether the NMDA/AMPA receptors-Ca²⁺ axis is responsible for increased ROS levels, we analyzed ROS levels in neurons treated with Aβ42 oligomers and exposed to memantine and CNQX to inhibit extracellular Ca²⁺ influx, or EGTA which reduces Ca²⁺ availability in the medium. We observed that both treatments resulted in lower levels of ROS (Fig. 4B), indicating that ROS increase is consequential to Ca²⁺ influx triggered by Aβ42 oligomers-mediated activation of AMPA and NMDA receptors, in agreement with previous reports (Fani et al, 2022).

Importantly, analyses of DDR in these same samples revealed a consistent decrease in DDR activation upon EGTA, Memantine and CNQX (Fig. 4C,D).

Overall, these results are consistent with a model in which the increase in ROS, caused by intracellular Ca²⁺ influx following the NMDA and AMPA receptors activation upon Aβ42 oligomers treatment, is responsible for DDR activation.

## Inhibition of Aβ42 oligomer-induced tDDR by tASOs decreases DDR activation and reduces Aβ42 oligomer toxicity

To determine the contribution of the newly observed Aβ42 oligomer-induced tDDR activation in neurons, we took advantage of tASOs, which allow the specific inhibition of telomeric DDR by targeting tdincRNAs generated at dysfunctional telomeres and necessary for a full activation of DDR at telomeres (Rossiello et al, 2017). We supplemented the culture medium of mouse hippocampal primary neurons with ASO vehicle (PBS), or tASOs (anti-teloG strand or anti-teloC strand) or a control ASO with an unrelated sequence before Aβ42 oligomers treatment. To determine their efficacy in inhibiting tDDR activation in neurons, we stained cells with antibodies against γH2AX, expected not to be significantly affected by tASOs as we previously demonstrated (Michelini et al, 2017; Rossiello et al, 2017), and against 53BP1 whose recruitment at dysfunctional telomeres depends on tdincRNA (Francia et al, 2012; Michelini et al, 2017; D'Alessandro et al, 2018; Pessina et al, 2019; Sharma et al, 2021). We observed that 53BP1 foci number was reduced in tASO-treated neurons (Fig. 5A), while γH2AX foci number remained unchanged (Fig. 5B) and control ASO had no impact on either DDR markers (Fig. 5A,B) this result becomes even more evident when proposed as the ratio of the number of 53BP1 foci over γH2AX foci per nucleus (Fig. 5C,D).

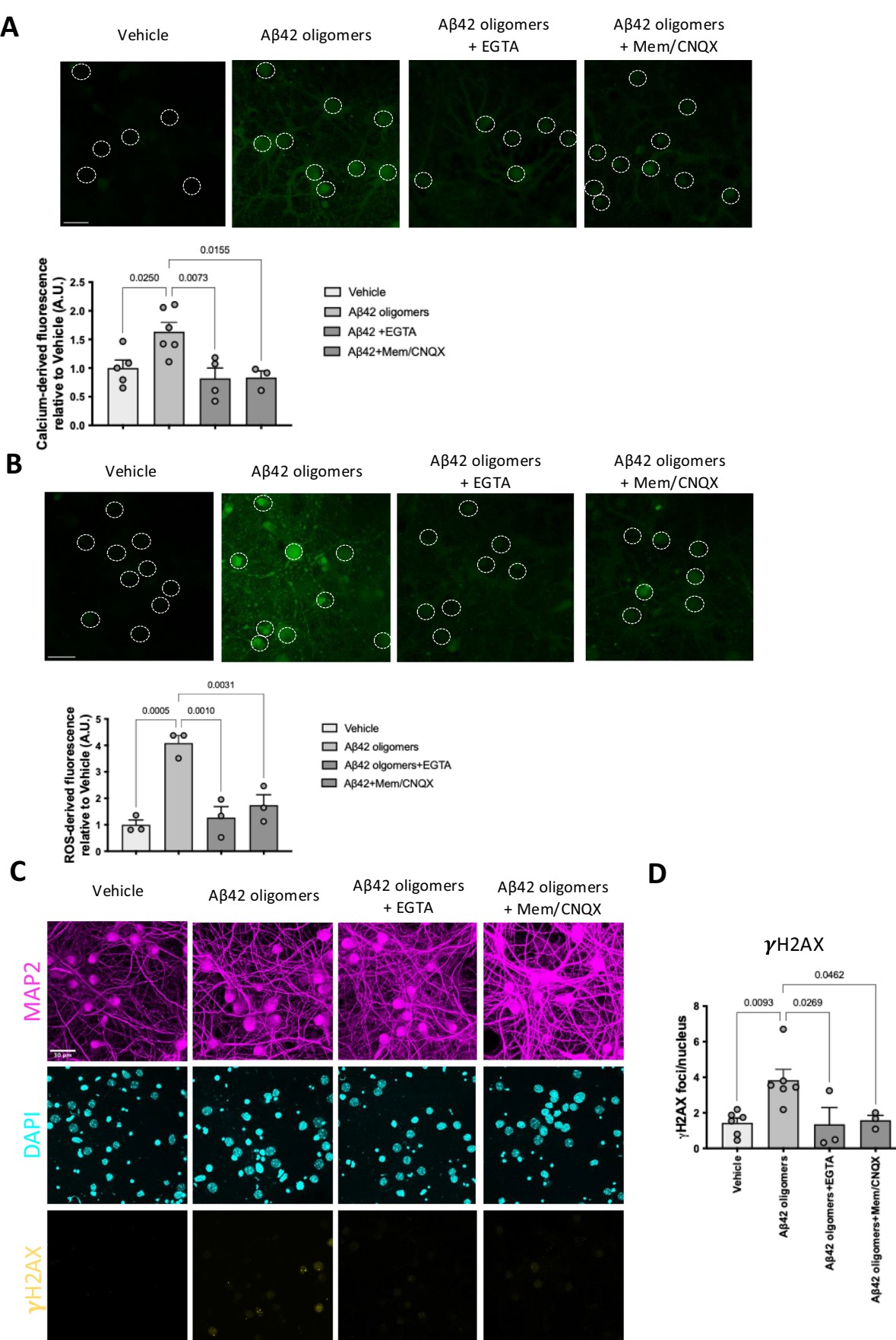

**Figure 4.  ROS-induced DDR upon synthetic Aβ42 oligomers administration in mouse hippocampal primary neurons depends on altered Ca²⁺ flux.**

(A–D) Primary neurons were treated for 1 h with 500 μM EGTA or 10 μM Memantine and 5 μM CNQX before treatment with Aβ42 oligomers or vehicle. Each circle represents an independent biological replicate. (A) Representative confocal microscopy images of mouse primary neurons incubated with Fluo-4 probe: green fluorescence is generated upon Ca²⁺ binding to the Fluo-4 probe. White dashed-line circles indicate the nuclei of neurons. Bar correspond to 20 μm The bar graph represents semiquantitative analysis of intracellular Ca²⁺-derived fluorescence. At least 40-50 cells for each condition in each replicate were analyzed. (Vehicle, $n = 5$; Aβ42 oligomers, $n = 6$; Aβ42 + EGTA, $n = 4$; Aβ42+Mem/CNQX, $n = 3$). (B) Representative confocal microscopy images of mouse primary neuron with CM-H₂DCFDA probe: the green fluorescence arises from oxidation of CM-H₂DCFDA probe. White dashed-line circles indicate the nuclei of neurons. The bar graph represents semiquantitative analysis of intracellular ROS intracellular ROS detected by CM-H₂DCFDA. White dashed-line circles indicate the nuclei of neurons. Bar correspond to 20 μm. At least 40-50 cells for each condition in each replicate were analyzed. (Vehicle, $n = 3$; Aβ42 oligomers, $n = 3$; Aβ42 + EGTA, $n = 3$; Aβ42+Mem/CNQX, $n = 3$). (C) Representative confocal images of immunofluorescence staining of mouse primary neurons with antibodies against MAP2 (magenta) as neuronal marker and γH2AX (yellow) as DDR marker, nuclei were stained with DAPI. (D) Bar graph represents the quantification of the average number of DDR foci per nucleus. At least 50 cells for each condition in each replicate were analyzed. (Vehicle, $n = 6$; Aβ42 oligomers, $n = 6$; Aβ42 + EGTA, $n = 3$; Aβ42+Mem/CNQX, $n = 3$). Ordinary one-way ANOVA was applied and data are represented as mean ± SEM. Source data are available online for this figure.

This here-demonstrated ability to inhibit Aβ-induced DDR at telomeres in mouse hippocampal primary neurons prompted us to test whether telomeric DDR contributes to Aβ42-induced toxicity in neurons. Thus, we treated mouse primary neurons with ASO vehicle (PBS) or control ASO or anti-teloG ASO before exposing them to Aβ42 oligomers—since anti-telo C (G-rich ASO) have in some instances been associated with some toxicity specifically in neuronal cells (Hagedorn et al, 2022), we pursued the use of the C-rich anti-teloG ASO only. Consistently with changes observed for DDR markers, p21 mRNA levels were reduced by tASO treatment (Fig. 6A) further indicating a reduction of DDR signaling upon tASO treatment. Next, we determined the impact of tDDR inhibition by tASO treatment on neurons viability using CellTiter-Glo luminescent cell viability assay kit based on ATP detection. We observed that Aβ42 oligomers reduced viability, and that tDDR inhibition restored viability to untreated levels (Fig. 6B), supporting the notion of a relevant role of tDDR in Aβ42 oligomers-mediated neurotoxicity. To further investigate the impact of tASO on apoptotic pathways, we analyzed by RT-qPCR Bax and Bcl-2 mRNA levels in the same cells. Consistent with our in vivo results (Figs. 1G and EV1D), we detected an increase in the Bax/Bcl2 ratio following the treatment with Aβ42 oligomers, which was significantly reduced by tDDR inhibition with tASO (Fig. 6C).

To further investigate whether tASO treatment impacts on the RNA molecular signatures of Aβ42 oligomers, we performed next-generation RNA-seq analyses in mouse primary neurons treated with control ASO and tASO before the treatment with Aβ42 oligomers. We observed that tASO treatment significatively deregulates 431 genes (DEGs) compared to control ASO (Fig. 6D). GO Biological processes analysis of these DEGs identified pathways implicated in neuronal synapsis and Ca²⁺ response (Fig. EV6A) suggesting that tDDR contributes to the pathways dysregulated by Aβ42 oligomers and that tASO treatment can reduce their toxic impact. To further investigate whether tASO treatment specifically impacts on genes induced by Aβ42 oligomers toxicity, we compared the 549 DEGs in Aβ42 oligomers versus vehicle with the 431 DEGs in Aβ42 oligomers Anti-TeloG versus Aβ42 oligomers control ASO samples. We identified 53 DEGs present in both conditions: of note, 41 of them are reverted by tASO treatment ("Reverted DEGs") (Fig. 6E), indicating that tASO can revert some of the biological pathways induced by Aβ42 oligomers. To confirm that, we performed a GO Biological processes analysis on "reverted DEGs" (Fig. 6F) which identified several neuronal processes, including Ca²⁺ and ROS-dependent processes that were reverted by tASO treatments. Noteworthy, tASO "reverted genes" were also found to

be dysregulated in the brain areas from AD patients compared to controls, as assessed through the Agora AD Knowledge Portal (Greenwood et al, 2020). Remarkably, among the reverted genes, 10 of them are consistently changed in their expression in post-mortem tissue of AD patients versus control, and we defined these as "concordant reverted DEGs" (Fig. 6G). Interestingly, they include genes involved in synaptic processes, such as Solute Carrier Family 17 member 6 (SLC17A6), implicated in glutamatergic transport across the membrane and neurotransmitter loading into the synaptic vesicles (Reimer, 2013; Cheret et al, 2021), and MDGA1, reported to modulate amyloid precursor protein action at the synapse (Kim et al, 2022).

In summary, we discovered that selective tDDR inhibition by tASO treatment is effective and leads to a reduction of apoptotic pathways and to the reversal of dysfunctional molecular signatures shared with AD in humans, and to increased neural viability.

## Inhibition of tDDR by tASOs upon Aβ42 oligomers treatment decreases DDR activation and rescues the Aβ42 oligomer-induced toxicity in human cortical neurons derived from iPSCs

To extend our conclusions to human neurons, we differentiated human cortical neurons from iPSCs, and we treated them with Aβ42 oligomers in the same manners of the mouse neurons. An immunofluorescence study of DDR activation using antibody against γH2AX and 53BP1, revealed an increased number of γH2AX and 53BP1 foci upon Aβ42 oligomers treatment (Fig. 7A,B), indicating that human neurons respond similarly to mouse ones. When we treated iPSC-derived human cortical neurons with ASO vehicle, or control ASO or tASOs prior to their exposure to Aβ42 oligomers, we observed a reduction of the number of 53BP1 foci upon tASOs treatment and unaltered γH2AX foci, demonstrating the contribution of telomeric DDR activation upon Aβ42 oligomers exposure (Fig. 7A–D) and demonstrating that tASO treatment turn off DDR also in human neuronal cell.

To determine the downstream effects of tDDR inhibition, we measured p21 mRNA levels and we observed its downregulation upon tDDR inhibition by tASO treatment (Fig. 7E). We next examined the effects of tASO on cell viability assessed by the use of CellTiter-Glo luminescent cell viability assay kit and we observed that Aβ42 oligomers reduced viability, and that tDDR inhibition restored viability to untreated levels in human neurons (Fig. 7F). Furthermore, tASO treatment reduced Bax/Bcl-2 ratio in human neurons, confirming that tASO has an impact on apoptotic genes (Fig. 7G).

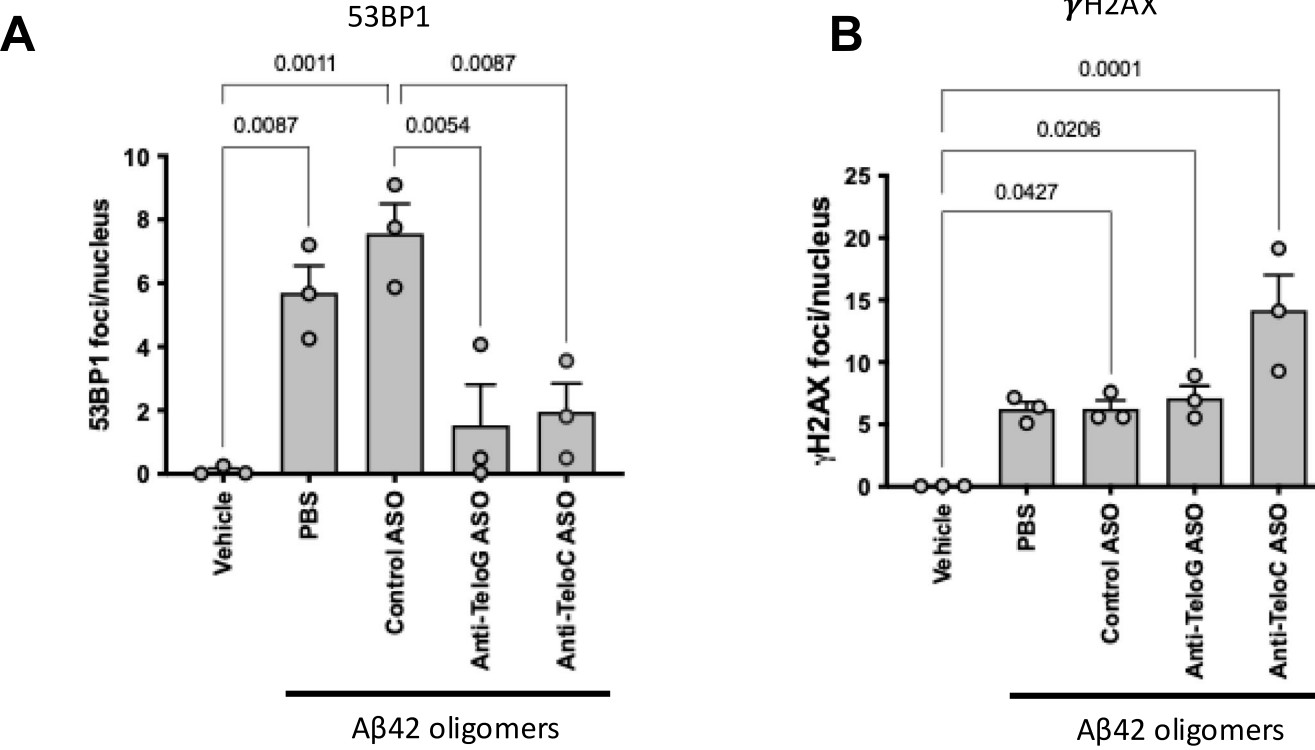

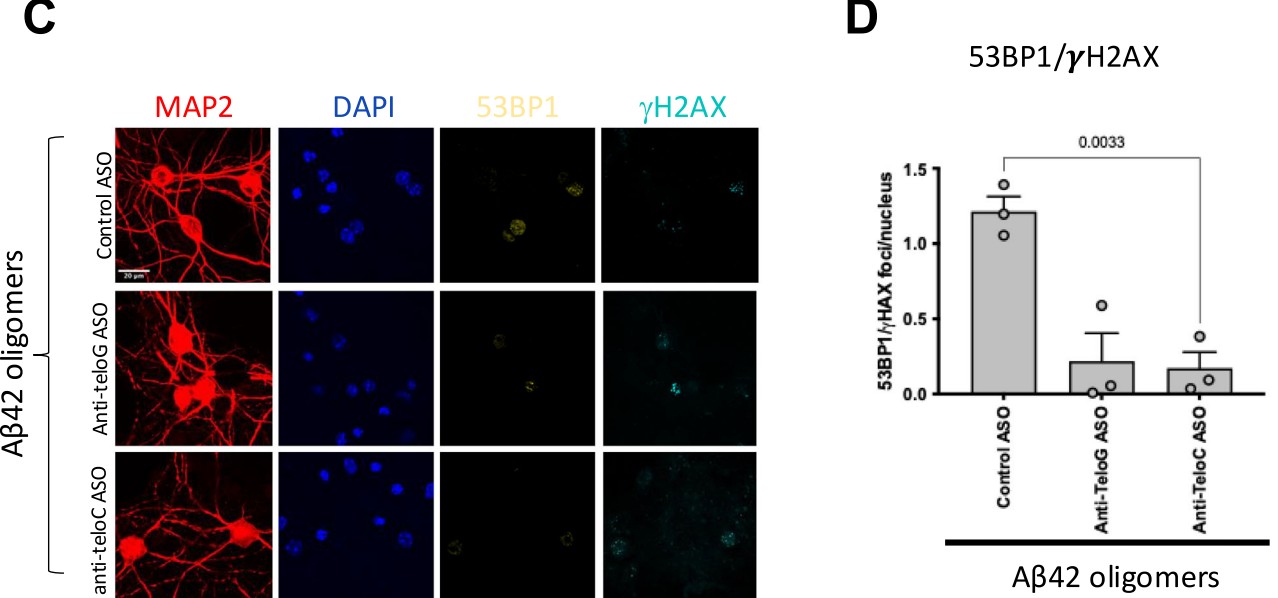

**Figure 5. Antisense oligonucleotides against tdincRNAs inhibit Aβ42 oligomers-induced DDR activation in mouse hippocampal primary neurons.**

(A–D) Mouse primary neurons were treated with ASO vehicle (PBS) or 10 µM anti-TeloG ASO or anti-Telo C ASO or Control ASO for 24 h before treatment with Aβ42 oligomers or vehicle. Each circle represents an independent biological replicate. (A, B) Bar graph represents the quantification of the average number of 53BP1 or γH2AX foci per nucleus, respectively. At least 50 cells for each condition in each replicate were analyzed. ($n = 3$ for each condition). (C) Representative confocal images of immunofluorescence staining of mouse primary neurons with antibodies against MAP2 (red) as neuronal marker, γH2AX (cyan) and 53BP1 (yellow) as DDR markers, nuclei were stained with DAPI. (D) Bar graph shows the ratio between the number of 53BP1 foci normalized on the number of γH2AX foci per nucleus per neuron. ($n = 3$ for each condition). Ordinary one-way ANOVA was applied and data are represented as mean ± SEM. Source data are available online for this figure.

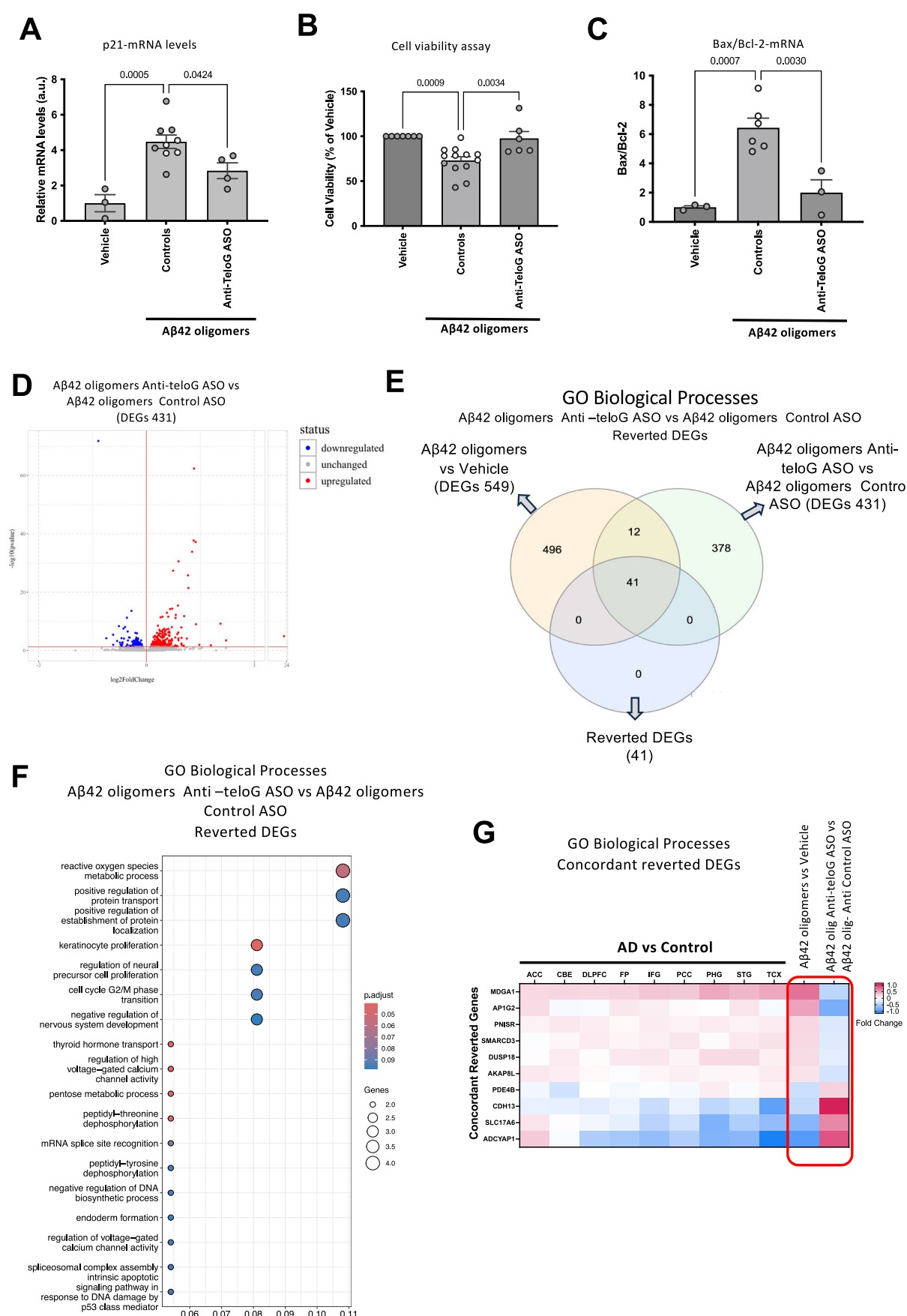

**A** p21-mRNA levels

**B** Cell viability assay

**C** Bax/Bcl-2-mRNA

**D** Aβ42 oligomers Anti-teloG ASO vs Aβ42 oligomers Control ASO (DEGs 431)

**E** GO Biological Processes
Aβ42 oligomers Anti –teloG ASO vs Aβ42 oligomers Control ASO
Reverted DEGs

Aβ42 oligomers vs Vehicle (DEGs 549)

Aβ42 oligomers Anti-teloG ASO vs Aβ42 oligomers Control ASO (DEGs 431)

Reverted DEGs (41)

**F** GO Biological Processes
Aβ42 oligomers Anti –teloG ASO vs Aβ42 oligomers Control ASO
Reverted DEGs

**G** GO Biological Processes
Concordant reverted DEGs

**Figure 6. Telomeric DDR inhibition reduces Aβ42 oligomers-induced neurotoxicity in mouse hippocampal primary neurons.**

(A–G) Mouse primary neurons were treated with ASO vehicle (PBS) or 10 μM ASO anti-Telo G or Control ASO for 24 h before treatment with Aβ42 oligomers or vehicle. Each circle represents an independent biological replicate (A) Bar graph shows the quantification of p21 mRNA levels relative to vehicle as measured by RT-qPCR. Controls showed as PBS (full dots) and control ASO (empty dots). (Vehicle, $n = 3$; Controls $= 9$, anti-TeloG, $n = 4$). (B) Bar graph represents cell viability as determined by CellTiter-Glo shown as a percentage of surviving cells compared to the vehicle. Controls are shown as PBS (full dots) and control ASO (empty dots). (Vehicle, $n = 7$; Controls $= 13$, anti-TeloG, $n = 6$). (C) Bar graph represents the quantification of the ratio between the mRNA levels of Bax and Bcl-2 relative to vehicle as measured by RT-qPCR. Controls showed as PBS (full dots) and control ASO (empty dots). (Vehicle, $n = 3$; Controls $= 6$, anti-TeloG, $n = 3$). (D) RNA-sequencing analysis was performed on $n = 3$ biological replicates for Aβ oligomers ASO Control and $n = 3$ replicates for Aβ 42 oligomers anti-teloG. Volcano plot showing significant differences in gene expression in primary neurons treated with Aβ42 oligomers Anti-telo G vs Aβ42 oligomers control ASO (differentially expressed genes (DEGs)). Each point represents a gene. The differential analysis was performed on R with DESeq2 (Love et al, 2014), differentially expressed genes were defined as genes with $p$ value adjusted (Benjamini–Hochberg) < 0.05 (genes with logFC >0 are considerate as upregulated, genes with logFC <0 as downregulated). (E) Venn diagram indicating DEGs common between Aβ42 oligomers vs vehicle (549) and Aβ42 oligomers anti-Telo G vs Aβ42 oligomers control ASO (431). 41 of these genes are reverted in the two conditions (Reverted DEGs). (F) The dotplot shows the significant Gene Ontology Biological processes from the enrichment analysis of "Reverted DEGs" performed with ClusterProfiler. The name of the pathway is reported on the left of the graph, the $x$-axis represents the gene ratio, dots size represents the number of differentially expressed genes present in the pathway and the color indicates the adjusted $p$ value of the pathway enrichment. Statistical analysis was performed with over-representation analysis for Gene Ontology Biological processes on DEGs, performed with ClusterProfiler (Wu et al, 2021). (G) Heatmap showing the nine reverted genes that are concordant with human AD patients vs control patients in the indicated brain areas (concordant reverted DEGs). ACC anterior cingulate cortex, CBE cerebellum, DLPFC dorsolateral prefrontal cortex, FP frontal pole, IFG inferior frontal gyrus, PCC posterior cingulate cortex, PHG parahippocampal gyrus, STG superior temporal gyrus, TCX temporal cortex. In (A–C), ordinary one-way ANOVA was applied and data are represented as mean ± SEM. Source data are available online for this figure.

These results demonstrate that tASO can mitigate the toxicity induced by Aβ42 oligomers in human cortical neurons.

## Discussion

AD is a complex and multifactorial condition in which multiple cellular pathways have been reported to undergo dysfunction, leading to a pathological phenotype (Wilson et al, 2023). Current therapeutic approaches approved worldwide aim to alleviate symptoms rather than slowing down disease progression and often come with significant side effects (Reisberg et al, 2003; Robinson and Keating, 2006). Three monoclonal antibodies have been approved by the American FDA, including aducanumab, lecanemab, and donanemab (Budd Haeberlein et al, 2022; Chen et al, 2024; CH et al, 2023; Sims et al, 2023), which target beta-amyloid products. Exploring alternative molecular pathways for more specific treatments is therefore crucial. Telomeric DDR accumulates with age and is known to play a role in age-related dysfunction (Fumagalli et al, 2012; Hewitt et al, 2012; Rossiello et al, 2022), making us speculate whether it could also be relevant in AD, an eminently age-related disease. Indeed, we observed an increased accumulation of DDR markers at telomeres in the brain of AD mouse model. Telomeric DDR was orthogonally validated by the increased production of tDDR-specific RNA (tdincRNA), which is essential for full DDR activation at telomeres. DDR activation was activated down to the most downstream elements, such as the CDK inhibitor p21, leading to the induction of a pro-apoptotic response. Similar and consistent conclusions were carried out in the isocortex and hippocampal formation of 3xTg-AD, two distinct and relevant brain areas, although we observed a generally less pronounced phenotype in the hippocampus, likely due to the higher susceptibility to neuronal cell death reported for this brain area in AD (Belfiore et al, 2019).

While our findings identified telomeric DNA damage, tDDR activation and altered gene expression in a mouse model known to recapitulate features of neuronal dysfunction and cognitive decline typical of AD (Belfiore et al, 2019), causality remains to be established, and this will be possible by tDDR inhibition in vivo.

In AD patients and mouse models, one of the earliest event that anticipates synaptic dysfunction and cognitive impairments is the rise of Aβ42 levels and the consequent accumulation of Aβ42 oligomers in the brain (Kayed and Lasagna-Reeves, 2013; Kayed et al, 2003; D'Andrea et al, 2001). For this reason, we adopted the model of mouse hippocampal primary neurons exposed to Aβ42 oligomers and examined the role of telomeric DDR, as these oligomers are known initiators of pathological events, impacting synaptic function, $Ca^{2+}$ homeostasis, redox state of neurons, and cell viability (Kayed et al, 2003; Fani et al, 2021; De Felice et al, 2007).

Our results indicate that the treatment with Aβ42 oligomers induces altered NMDAR/AMPAR-mediated $Ca^{2+}$ flux, ROS production, and consequent tDDR activation. This is associated with gene expression alterations, as determined by transcriptomics analyses, which are akin to gene expression alterations observed in human AD brains: this supports the notion that this model recapitulates Aβ oligomer-mediated AD dysfunction in neurons and it is suitable to investigate pathogenic mechanisms, including DDR activation, particularly at telomeres.

To determine the contribution of tDDR, we inhibited its activation using a set of tools, tASOs, that we previously validated in cultured cells and in vivo in mice (Michelini et al, 2017; Rossiello et al, 2017). tASOs, by targeting tdincRNA, inhibit tDDR activation and contrast the effects of its persistent activation as demonstrated by a decrease in 53BP1 foci formation and downregulated p21 expression. Such tDDR inhibition is beneficial for neurons when exposed to Aβ42 oligomers, as demonstrated by normalized Bax and Bcl2 apoptotic genes levels, and improved cell viability. These findings are supported by transcriptomics analyses, which demonstrate that tASO-mediated tDDR inhibition rescues gene expression dysregulated by Aβ42 oligomers. Notably, tDDR inhibition by tASOs is efficacious also in human cortical neurons. Of note, we confirmed the efficacy of tASO in inhibiting tDDR and apoptosis induced by Aβ42 oligomers in human cortical neurons derived from iPSCs, leading to a rescue of vitality, also in human cells.

In conclusion, by proposing a mechanism for telomeric DNA damage generation and tDDR pathway activation induced by Aβ42

oligomers and mediated by an influx of $Ca^{2+}$ ions through AMPA and NMDA receptors and subsequent ROS generation, we suggest a model to improve our understanding of Aβ42 oligomer-induced cellular toxicity (Fig. 8). The study of the effects of tDDR inhibition in mouse and human neurons lays the groundwork for exploring new therapeutic approaches to mitigate pathological events in AD.

# Methods

## Methods and protocols

### Animals

Experiments involving animals were performed in accordance with the Italian Laws (D.lgs. 26/2014), which enforce the Directive 2010/63/EU (Directive 2010/63/EU of the European Parliament and of the Council of 22 September 2010 on the protection of animals used for scientific purposes). Accordingly, the project has been authorized (n. 1089/2015-PR) by the Italian Competent Authority (Ministry of Health). 3XTg-AD female mice homozygous for all three mutant alleles (homozygous for the *Psen1* mutation and homozygous for the co-injected APPSwe and tauP301L transgenes (Tg (APPSwe,tauP301L)1Lfa/Mmjax) were purchased from Jackson Laboratory. They were bred and housed in the IIT animal facility in Genoa. To produce pups used to make mouse primary neurons, C57BL/6J mice were purchased from Charles River, Italy. Those animals were bred and housed in the IFOM animal facility.

### Cells and treatment

Mouse hippocampal primary neurons. Primary cultures of hippocampal and cortical neurons established from postnatal day 0–2 pups were plated on dishes or glass coverslips coated with poly-D-lysine hydrobromide (Merck). Cultures were used for experiments after 14 days in vitro (DIV) and incubated at 37 °C for all treatments in Neurobasal medium (Thermo Fisher Scientific) supplemented with B27 (Thermo Fisher Scientific), Glutamax (Thermo Fisher Scientific) and Penicillin/Streptomycin (Euroclone). The neurons were treated with Aβ oligomers (10 μM) or vehicle for 48 h in all experiments involving Aβ oligomer treatments. For the $Ca^{2+}$ deprivation experiment, ethylene glycol-bis(2-aminoethylether)-N,N,N′,N′-tetraacetic acid (EGTA 500 μM, Merck) was added to the medium before Aβ oligomer treatment. For NMDA/AMPA receptor inhibition, mouse primary neurons were treated for 60 min with or without 5 μM CNQX (AMPAr

**Reagents and tools table**

| Reagent/resource | Reference or source | Identifier or catalog number |
| --- | --- | --- |
| **Experimental models** | | |
| 3xTg-AD (*M. musculus*) | The Jackson Laboratory | Strain #:033930 |
| C57BL/6J (*M. musculus*) | Charles River | N/A |
| Hippocampal mouse primary neurons | https://www.nature.com/articles/nprot.2012.099 | N/A |
| HT-22 cells (*M. musculus*) | Salk Institute | N/A |
| Induced pluripotent stem cells (*H. sapiens*) | HipSci (ECACC) | HPSI0613i-nukw_1 |
| **Antibodies** | | |
| Anti-β-amyloid 6E10 | Biolegend | 803001 |
| Oligomer A11 | Thermo Fisher Scientific | AHB0052 |
| Chicken anti-MAP2 | Abcam | ab92434 |
| Rabbit anti-phospho-histone H2A.X (Ser139) (20E3) | Cell Signaling | 9718 |
| Mouse anti-phospho-histone H2A.X (Ser139) | Millipore | 05-636-I |
| Rabbit anti-53BP1 (H-300) | Santa Cruz | sc-22760 |
| Rabbit anti-53BP1 | Bhetyl | A300-272A |
| Mouse anti-phospho-ATM (Ser1981) clone 10H11.E12 | Millipore | 05-740 |
| Rabbit anti-phospho-(Ser/Thr) ATM/ATR substrate | Cell Signaling | 2851L |
| Goat Anti-Mouse IgG (H + L) HRP Conjugate | Biorad | 1706516 |
| Cy3 D/M | Jackson | 715-165-150 |
| Cy3 D/R | Jackson | 711-165-152 |
| A647 D/M | Thermo Fisher Scientific | A31571 |
| A647 D/R | Thermo Fisher Scientific | A31573 |

| Reagent/resource | Reference or source | Identifier or catalog number |
|---|---|---|
| Alexa Fluor 488 goat anti-chicken | Thermo Fisher Scientific | A11039 |
| Rabbit anti-Bax | MBL international | JM-3032-100 |
| Mouse anti-Bcl2 | Thermo Fisher Scientific | 13-8800 |
| Human-ANTI-Centromere (Kinetochore) | Antibodies Incorporated | 9101-02 |
| **Oligonucleotides and other sequence-based reagents** | | |
| PCR primers | This study | Table 1 |
| **Chemicals, enzymes, and other reagents** | | |
| Poly-D-Lysine hydrobromide | Merck | P6407 |
| Neurobasal | Thermo Fisher Scientific | 21103049 |
| B27 Supplement | Thermo Fisher Scientific | 17504044 |
| Glutamax | Thermo Fisher Scientific | 35050061 |
| Penicillin/streptomycin | Euroclone | ECB3001L |
| DMEM high glucose | Euroclone | ECB7501L |
| Fetal bovine serum | Merck | F7524 |
| L-Glutamine | Euroclone | ECB3000D |
| Abeta(1-42) chemical peptide | GenScript | SC1208 PepPowerTM Chemical Peptide - DAEFRHDSGYEVHHQKLVFFAEDVGSNKGAIIGLMVGGVVI A |
| Control peptide | GenScript | KVKGLIDGAHIGDLVYEFMDSNSAIFREGVGAGHVHVA QVEF |
| Ethylene glycol-bis(2-aminoethylether)-N,N,N',N'-tetraacetic acid (EGTA) | Merck | E3889 |
| CNQX disodium salt hydrate | Merck | C239 |
| Memantine hydrichloride | Merck | M9292 |
| N-acetyl-L-cysteine | Merck | A9165 |
| Phosphate BS | | |
| Control ASO | Qiagen | ACTGATAGGGAGTGGTAAACT |
| Anti-Telo G ASO | Qiagen | CCCTAACCCTAACCCTAACCC |
| Anti-Telo C ASO | Qiagen | GGGTTAGGGTTAGGGTTAGGG |
| Matrigel | Corning | 354230 |
| mTeSR Plus Kit | Stemcell Technologies | 05825 |
| Hepes | Euroclone | ECM0180L |
| Bolt™ Bis-Tris Plus Mini Protein Gels, 4–12% | Thermo Fisher Scientific | NW04120B0X |
| Amersham TM ProtranTM 0.45 μM NC nitrocellulose blotting membrane | Cytiva | 10600002 |
| SuperSignal™ Pierce™ West Pico PLUS Chemiluminescent substrate | Thermo Fisher Scientific | 34580 |
| Paraformaldehyde, 4% in PBS | Himedia | TCL119 |
| 4',6-Diamidino-2-phenylindole dihydrochloride (DAPI) | Merck | D9542 |
| Albumin bovine BSA | SEQENS | 1005-70 |
| Gelatin from cold water fish skin | Sigma-Aldrich | G7765 |
| Cy5-conjugated TelC PNA probe | Panagene | F1003 |
| Triton X 100 | Merck | T8787 |
| Tween 20 | Panreac Applichem | A4974 |
| Mowiol | Merck | 81381 |
| Glycerol | Carlo Erba | 453752 |

| Reagent/resource | Reference or source | Identifier or catalog number |
|---|---|---|
| VECTASHIELD® Antifade Mounting Medium | Vector Laboratories | H-1000-10 |
| Maxwell® RSC simplyRNA Tissue Kit | Promega | AS134 |
| SuperScript VILO cDNA Synthesis Kit | Thermo Fisher Scientific | 11754050 |
| LightCycler 480 SYBR Green I Master | Roche | 4707516001 |
| TURBO DNase | Thermo Fisher Scientific | AM2238 |
| SuperScript™ III Reverse Transcriptase | Thermo Fisher Scientific | 18080044 |
| MicroSpin™ G-50 columns | Cytiva | 27533002 |
| CellTiter-Glo.2 Luminescent Cell Viability Assay | Promega | G9242 |
| Ionomycin calcium salt | Merck | I3909-1ML |
| Fluo-4 AM | Themo Fisher Scientific | F14201 |
| Hydrogen peroxide solution | Merck | 216763 |
| 5-(and-6)-chloromethyl-2′,7′-dichlorodihydrofluorescein diacetate (CM-H2DCFDA) | Themo Fisher Scientific | D23844 |
| Illumina stranded mRNA prep ligation kit | Illumina | 20040534 |
| **Software** | | |
| R | v.4.4.0 | |
| Leica LAS X Software | | |
| GraphPad Prism | v.10.0 | |
| Cell Profiler | v4.2.6 | |
| Fiji/ImageJ | V2.9 | |
| **Other** | | |
| Leica SP5 confocal microscope | Leica | |
| Maxwell® RSC Instrument | Promega | |
| NanoVueTM Plus Spectrophotometer | GE Healthcare | |
| LightCycler 96 Roche | Roche | |
| EnVision Multilabel Plate Reader | Perkin Elmer | |
| Leica SP8 confocal microscope | Leica | |
| nf-core rnaseq pipeline | | v3.12.0-g3bec233 |
| Platform | | x86_64-apple-darwin20 |
| ChemiDoc MP Imaging System | Biorad | |
| AlzCode Web Resource | http://www.alzcode.xyz | |
| Agora AD Knowledge portal | https://agora.adknowledgeportal.org/ | |
| Qubit Fluorimeter | Thermo Fisher Scientific | |
| Agilent Bioanalyzer 2100 instrument | Agilent Technologies | |
| Illumina NextSeq550Dx sequencer | Illumina | |

blocker, Merck) and 10 µM memantine (NMDAr blocker, Merck) before Aβ oligomers treatment.

For the ROS experiment, mouse primary neurons were treated with 1 mM N-acetyl-L-cysteine (Merck) for 30 min before Aβ oligomer treatment. For telomeric DDR inhibition, neurons were treated for 24 h with ASO vehicle (PBS), control ASO, anti-Telo G and anti-Telo C (10 µM, Qiagen) ASO, followed by the addition of Aβ oligomers (10 µM) or vehicle. Cells were processed after 48 h for immunofluorescence, RNA extraction, and viability assay.

HT-22. The mouse hippocampal HT-22 cell line (Salk Institute) was maintained in Dulbecco's modified Eagle medium (DMEM, Thermo Fisher Scientific) containing 10% fetal bovine serum (FBS, Thermo Fisher Scientific), 2 mM L-Glutamine (Euroclone) and 1% penicillin/streptomycin (Euroclone). The cells were grown in a humidified incubator (5% CO₂, 37 °C) and passaged when reaching 70% confluency. The cells were seeded overnight prior to treatment with Aβ42 oligomers (5 µM) or vehicle for 24 h in all experiments involving Aβ42 oligomer treatment. Cells were routinely

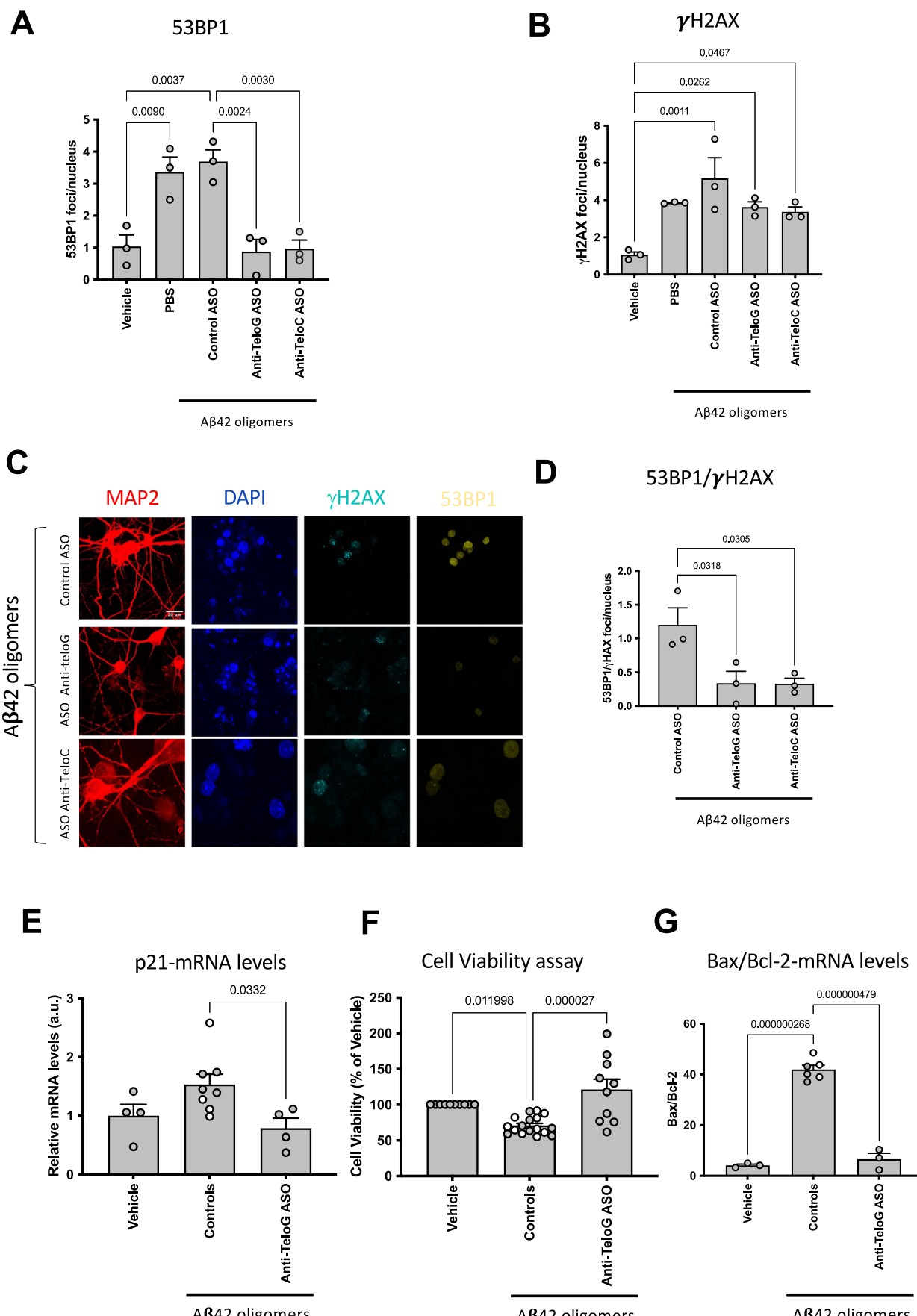

**Figure 7.  Antisense oligonucleotides against tdincRNAs inhibit DDR activation induced by Aβ42 oligomers and reduce Aβ42 oligomer-induced neurotoxicity in human cortical neurons derived from iPSCs.**

(**A–G**) Human cortical neurons derived from iPSCs were treated with ASO vehicle (PBS) or 10 µM ASO anti-Telo G or ASO anti-Telo C or ASO control for 24 h before treatment with Aβ42 oligomers or vehicle. Each circle represents an independent biological replicate. (**A–D**) DDR analyses. (**A, B**) Bar graph represents the quantification of the average number of 53BP1 or γH2AX foci per nucleus, respectively. At least 50 cells for each condition in each replicate were analyzed. ($n = 3$ for each condition). (**C**) Representative confocal images of immunofluorescence staining of mouse primary neurons with antibodies against MAP2 (red) as neuronal marker, γH2AX (cyan) and 53BP1 (yellow) as DDR markers, nuclei were stained with DAPI. (**D**) Bar graph represents the quantification of ratio between 53BP1 foci normalized on the numbers of γH2AX foci per nucleus in each neuron. At least 50 cells for each condition in each replicate were analyzed. ($n = 3$ for each condition). (**E**) Bar graph shows the quantification of p21 mRNA levels relative to vehicle as measured by RT-qPCR. Controls showed as PBS (full dots) and control ASO (empty dots). (Vehicle, $n = 4$; Controls $= 8$, anti-TeloG, $n = 4$). (**F**) Bar graph represents cell viability as determined by CellTiter-Glo shown as percentage of surviving cells compared to vehicle. Controls are shown as PBS (full dots) and control ASO (empty dots). (Vehicle, $n = 10$; Controls $= 17$, anti-TeloG, $n = 10$). (**G**) Bar graph represents the quantification of the ratio between the mRNA levels of Bax and Bcl-2 relative to vehicle as measured by RT-qPCR. Controls are shown as PBS (full dots) and control ASO (empty dots). Vehicle, $n = 3$; Controls $= 6$, anti-TeloG, $n = 3$) Ordinary one-way ANOVA was applied and data are represented as mean ± SEM. Source data are available online for this figure.

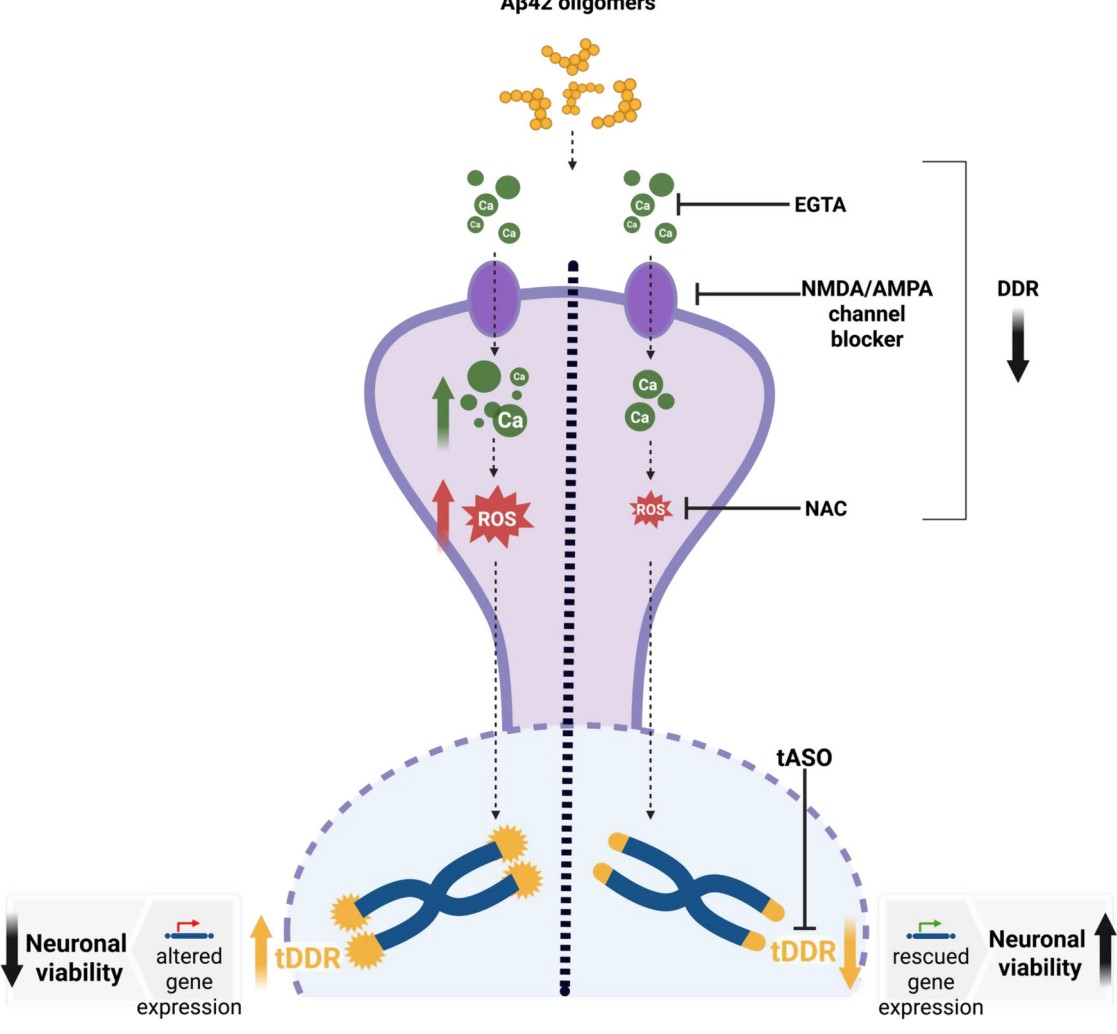

**Figure 8.  Proposed model for tDDR generation and impact of its inhibition by tASO in neurons.**

Left side of the scheme: Aβ42 oligomers induce increased intracellular Ca²⁺ levels mediated by NMDA/AMPA receptors. This leads to a rise in ROS, causing telomeric DDR (tDDR), which is associated with reduced neuronal viability. Right side of the scheme: inhibition of Ca²⁺ availability in the culture medium by EGTA, Ca²⁺ influx by NMDA/AMPA blockers, or ROS generation by NAC, all cause decreased tDDR activation. Inhibition of tDDR by tASO decreases tDDR activation, normalizes gene expression, and increases neuronal viability. Dashed arrows represent the functional consequences of the events described in the graph. Created with BioRender.com.

authenticated by STR profiling and tested for mycoplasma contamination by the IFOM cell culture facility.

Human cortical neurons derived from iPSCs. Human iPSCs were purchased from HipSci (HPSI0613i_nukw_1) and cultured as colonies on Matrigel hESC-qualified (Corning)-coated plates in mTeSR-plus (Stemcell Technologies).

For the differentiation, the Studer's protocol was applied as described in Qi et al (2017).

### Aβ42 oligomer preparation

Aβ$_{1-42}$ (Aβ42 oligomers) oligomers were obtained from a depsi-peptide Aβ$_{1-42}$, which was purchased from GenScript. At variance with the native peptide, the depsi-peptide is highly soluble and it has a much lower propensity to aggregate, thus preventing the spontaneous formation of seeds in the solution (Beeg et al, 2011; Balducci et al, 2010). The native Aβ$_{1-42}$ peptide was then obtained from the depsi-peptide by a "switching" procedure in basic conditions. The alkaline stock solution (400 μM) was diluted to 100 μM Aβ in PBS and incubated for 18–20 h at 4 °C. Aβ$_{1-42}$ preparations were diluted in the medium to reach the final concentration (10 μM). Before adding oligomers to the medium, Hepes 1 M (Euroclone) was add to the medium to reach the final concentration (80 μM). To confirm the state of the Aβ$_{1-42}$ peptide, we performed Western blot analysis. Briefly, the oligomers solution was subjected to 4–12% Bis-Tris Gel Separation (Thermo Fisher Scientific) electrophoresis and transferred to nitrocellulose membranes (Cytiva). These membranes were incubated with a primary antibody against mouse monoclonal human Aβ 6E10 and oligomers A11. For detection, the membrane was incubated with a Goat Anti-Mouse IgG (H + L) HRP Conjugate antibody (Biorad). Immunoreaction signals were visualized with SuperSignal™ Pierce™ West Pico PLUS (Thermo Fisher Scientific). The gel for Comassie-based staining was stained with InstantBlue solution (ISB1L, Expedeon).

### Immunofluorescence on mouse and human neurons

Cells were fixed with 4% PFA. After permeabilization with Triton 0.2% (Merck) and incubation with blocking solution (5% bovine serum albumin and 2% Gelatin from cold water), cells were stained with primary antibody O.N. at 4 °C, brought to RT, washed and incubated with secondary antibodies for 40 min at RT. Nuclei were stained with 4,6-diamidino-2-phenyl-lindole (DAPI; 1 mg ml$^{-1}$, Merck). Samples were mounted in mowiol (Merck). Image acquisition was performed in a Leica SP5 confocal microscope. The detection parameters were set in the control samples and were kept constant across specimens.

### ImmunoFISH on neurons

Mouse primary neurons were treated as described (https://bio-protocol.org/e999) with some adaptation. Cells were denatured at 80 °C for 5 min and incubated with Cy5-conjugated TelC PNA probe (F1003—Panagene). After washes, immunofluorescence was performed. Image acquisition was performed in a Leica SP5 confocal microscope. The detection parameters were set in the control samples and were kept constant across specimens.

### Immunofluorescence on brain tissue

A measure of 4-mm tissue sections were fixed for 10 min in 4% PFA, incubated in blocking solution (2% BSA, 0.1% Tween in PBS) for 1 h at RT. Then sections were incubated for 1 h at RT with primary antibodies, washed in blocking solution and incubated for 1 h at RT with secondary antibody. Nuclei were stained with DAPI (1 mg ml$^{-1}$). Samples were mounted with glycerol solution. Image acquisition was performed in a Leica TCS SP5 confocal microscope. The detection parameters were set in the control samples and were kept constant across specimens.

### ImmunoFISH on brain tissue

Brain sections were treated as described (https://en-cdn.bio-protocol.org/pdf/Bio-protocol1000) with some adaptations. Sections were denatured at 80 °C for 5 min and incubated with Cy5-conjugated TelC PNA probe (Panagene). After washes, immunofluorescence was performed. Samples were mounted in VECTASHIELD® Antifade Mounting Medium (Vector Laboratories). Image acquisition was performed in a Leica SP5 confocal microscope. The detection parameters were set in the control samples and were kept constant across specimens.

### RNA extraction

RNA extraction from mouse primary neurons, human cortical neurons and mouse brains total RNA was performed using Maxwell® RSC Instrument (Promega), with Maxwell® RSC simplyRNA Tissue Kit (Promega), following the manufacturer's instructions. NanoVue™ Plus Spectrophotometer (GE Healthcare) was used to detect RNA quantity and purity. RNA purity was ascertained via NanoVue 260/280 and 260/230 ratios.

### Reverse transcription quantitative PCR

For primary neurons, human cortical neurons and mouse brains, 1–2 μg of total cell RNA was reverse transcribed into cDNA using SuperScript VILO cDNA Synthesis Kit (Thermo Fisher Scientific). A volume corresponding to 25–50 ng of cDNA was used for each RT-qPCR reaction using Roche LightCycler 480 SYBR Green I Master (Roche) and the reaction was run on LightCycler 96 Roche detection system. Each reaction was performed in triplicate. Primer sequences (5′–3′ orientation) are reported in Table 1.

### Strand-specific RT-qPCR for tdincRNA detection

Detection of tdincRNAs was performed as previously described (Rossiello et al, 2017), with some modifications. Briefly, RNA samples were treated with TURBO DNase (Thermo Fisher Scientific) at 37 °C for 1 h. Next, 1 μg of total RNA was reverse transcribed using the SuperScript™ III Reverse Transcriptase kit (Thermo Fisher Scientific) with strand-specific primers. cDNA was passed on a MicroSpin™ G-50 columns (Cytiva) and qPCR was performed using Roche LightCycler 480 SYBR Green I Master Mix (Roche) and run on LightCycler 96 Roche detection system. A volume of cDNA corresponding to 20 ng of initial RNA was used. Each reaction was performed in triplicate. Rplp0 was used as a control gene for normalization. Primer sequences (5'–3' orientation) are reported in Table 1.

### Cell viability

Neuronal viability was evaluated by ATP detection (CellTiter-Glo.2 Luminescent Cell Viability Assay; Promega) accordingly to the manufacturer instructions. Briefly cell were plated on 96-well plate and were lysed and incubated with CellTiter-Glo® 2.0 Reagent for 10' and luminescence was recorded with EnVision multilabel plate reader (Perkin Elmer).

**Table 1. Primer sequences.**

| Primer name | Sequence (5′ → 3′) |
|---|---|
| **Primers for RT-qPCR** | |
| Mouse p21 Fw | TTGCCAGCAGAATAAAAGGTG |
| Mouse p21 RV | TTTGCTCCTGTGCGGAAC |
| Mouse Bax Fw | CTGAGCTGACCTTGGAGC |
| Mouse Bax RV | GACTCCAGCCACAAAGATG |
| Mouse BCL2 Fw | TGAAGCGGTCCGGTGGATAC |
| Mouse BCL2 Rv | GGCGGCGGCAGATGAATTAC |
| Human p21 Fw | GGTGGACCTGGAGACTCTCAG |
| Human p21 Rv | TCCTCTTGGAGAAGATCAGCCG |
| Human Bax Fw | AGCAAACTGGTGCTCAAGG |
| Human Bax RV | TCTTGGATCCAGCCCAAC |
| Human BCL2 Fw | AGTACCTGAACCGGCACCT |
| Human BCL2 Rv | GCCGTACAGTTCCACAAAGG |
| **Housekeeping genes for human cells** | |
| Rplp0 Fw | TTCATTGTGGGAGCAGAC |
| Rplp0 Rv | CAGCAGTTTCTCCAGAGC |
| **House keeping gene for mouse** | |
| Rplp13Fw | ATCCCTCCACCCTATGACAA |
| Rplp13Rv | GCCCCAGGTAAGCAAACTT |
| **qPCR primer sequences for tdincRNA detection** | |
| Rplp0 Fw | TTCATTGTGGGAGCAGAC |
| Rplp0 Rv | CAGCAGTTTCTCCAGAGC |
| teloC Rv | CCCTAACCCTAACCCTAA |
| teloG Rv | GGGTTAGGGTTAGGGTTA |
| **RT primer sequences for tdincRNA detection** | |
| telo Fw | CGGTTTGTTTGGGTTTGGGTTTGGGTTTGGGTTTGGGTT |
| telo Rv | GGCTTGCCTTACCCTTACCCTTACCC TTACCCTTACCCT |

## Ca²⁺ measurements

Cytosolic $Ca^{2+}$ levels were measured in living mouse hippocampal primary neurons after the different treatments, or after adding 1 μM ionomycin (Merck) for 1 h as a positive control. The cells were then washed with PBS and loaded with 2 μM Fluo-4 AM (Thermo Fisher Scientific) for 10 min. Image acquisition was performed in a Leica SP8 confocal microscope. The detection parameters were set in the control samples and kept constant across specimens.

## ROS measurements

ROS levels were measured in living mouse hippocampal primary neurons after the different treatments, or after adding 400 μM $H_2O_2$ (Merck) for 1 h, as a positive control, and then by loading 5 μM 5-(and-6)-chloromethyl-2′,7′-dichlorodihydrofluorescein diacetate (CM-$H_2$DCFDA, Thermo Fisher Scientific) in the last 15 min of the different treatments. Image acquisition was performed in a Leica SP8 confocal microscope. The detection parameters were set in the control samples and kept constant across specimens.

## Illumina stranded mRNA sequencing

RNA-sequencing analysis was performed on three biological replicates for Aβ oligomers ASO Control, two replicates for Aβ samples, three replicates for Aβ 42 oligomers anti-teloG and two replicates for Vehicle conditions. The abundance and integrity of total RNA was assessed using Qubit fluorimeter and the Agilent Bioanalyzer 2100 instrument (Agilent Technologies). For each sample, 500 ng of total RNA were used to generate a library of fragments using Illumina Stranded mRNA prep ligation kit. Oligo(dT) magnetic beads purify and capture the mRNA molecules containing polyA tails. The purified mRNA was fragmented and copied into first strand complementary DNA (cDNA) using reverse transcriptase and random primers. In a second strand cDNA synthesis step, dUTP replaces dTTP to achieve strand specificity. The final steps added adenine (A) and thymine (T) bases to fragment ends and ligate adapters. The resulting products were purified and selectively amplified with 12 cycles of PCR to generate an indexed library of fragments. The library was quantified using Qubit 4.0 fluorimeter, checked on Agilent Bioanalyzer 2100 then loaded on Illumina NextSeq550Dx sequencer for sequencing, following the manufacturer's instructions. Library fragments were sequenced using 2x75nt readmode, thus sequencing 75 nucleotides from both ends of each fragment; on average, ~50 Million Paired-end fragments were sequenced for each sample. Sequencing results were generated in fastq.gz format.

## RNA-seq processing and analysis

Reads were aligned, on mm10 genome, and count with the NF-core/rnaseq pipeline (Mus Musculus bimaculoides genome assembly; Ewels et al, 2020) (v3.12.0-g3bec233). The quality check was performed with FastQC (Andrews, 2010). The differential analysis was performed on R with DESeq2 (Love et al, 2014), differentially expressed genes were defined as genes with $p$ value adjusted (Benjamini–Hochberg) < 0.05 (genes with logFC > 0 are considerate as upregulated, genes with logFC < 0 as downregulated). VolcanoPlot were done with ggplot2 (Create Elegant Data Visualizations Using the Grammar of Graphics • ggplot2) and ggbreak (Xu et al, 2021). Over-representation analysis for Gene Ontology Biological Processes on DEGs was performed with ClusterProfiler (Wu et al, 2021).

Significant association of DEGs with AD genes was obtained performing "Gene Set Evaluation" on AlzCode platform (Lin et al, 2022) (http://www.alzcode.xyz). Specifically, DEGs in Ab42 oligomers vs vehicle were submitted as input genes, AlzGene dataset was used as known AD-associated genes and integrated PPI was chosen as molecular interaction network to assess how significantly the queried gene set interacts with the list of AD-associated genes compared to random baseline. The significance is calculated by permutation test (Lin et al, 2022).

Agora AD Knowledge portal was also used to compare the expression of genes of interest in our experimental set up with the expression of the same genes in different brain areas from AD affected patients versus healthy controls (Greenwood et al, 2020). Specifically, human orthologues of genes of interest were submitted in the gene comparison tool and expression data for the same genes in Anterior Cingulate Cortex, Cerebellum, Dorsolateral Prefrontal Cortex, Frontal Pole, Inferior Frontal Gyrus, Posterior Cingulate Cortex, Parahippocampal gyrus, Superior Temporal Gyrus, Temporal Cortex was downloaded. The fold change was plotted on GraphPad Prism as a heatmap. Concordant genes were defined as genes that show a trend of dysregulation that is concordant with their expression in ≥5 human brain areas from post-mortem tissue of AD patients versus control (meaning either concordantly upregulated or downregulated in AD brains and in our condition of interest).

### Statistical analyses

All experiments were performed at least three times. For in vivo experiments, 3–10 mice were analyzed per condition. All bar graph results are shown as mean ± s.e.m. or s.d. as indicated. No statistical methods were used to predetermine sample size, but our sample sizes are similar to those reported in previous publications. Data distribution was assumed to be normal, but this was not formally tested for all analyses. No data were excluded from analyses. Data collection and analysis were not performed blind to the conditions of the experiments. No randomization method was used to allocate animals or conditions in vitro to experimental groups. $P$ values were calculated by the indicated statistical tests using either R (v.3.6.0), or GraphPad Prism (v.10.0) software. In figure legends, the number of animals or biological replicates is reported as single dots in the graph. Specifically, biological replicates refer to samples obtained from independent biological experiments, meaning were derived from different cultures of mouse primary neurons, each originating from distinct pools of pups. Additionally, different solutions of Aβ oligomers or ASO were used in these independent experiments. The number of cells analyzed is indicated in the figure legends. Unpaired $t$ test was applied in Figs. 1A–H and EV1A–E; Unpaired $t$ test was applied in Figs. 2A–F and EV3A–C. Permutation test (Lin et al, 2022) as applied in Fig. EV4B Ordinary one-way ANOVA with Šidák's post hoc test was applied in Fig. 3B. Ordinary one-way ANOVA with Multiple comparison post hoc test in Fig. 3C. Ordinary one-way ANOVA with Tukey's post hoc test was applied in Fig. 3E. Ordinary one-way ANOVA with Tukey's post hoc test was applied in Fig. 3F. Ordinary one-way ANOVA with Tukey's post hoc test and with Šidák's post hoc test was applied in Fig. 3G. Ordinary one-way ANOVA with Tukey's post hoc test was applied in Fig. 4A. Ordinary one-way ANOVA with Tukey's post hoc test and with Šidák's post hoc test was applied in Fig. 4B. Ordinary one-way ANOVA with Tukey's post hoc test was applied in Fig. 4D. Ordinary one-way ANOVA with Tukey's post hoc test was applied in Fig. 5A. Ordinary one-way ANOVA with Šidák's post hoc test was applied in Fig. 5B. Ordinary one-way ANOVA with Tukey's post hoc test was applied in Fig. 5D. Ordinary one-way ANOVA with Šidák's post hoc test was applied in Fig. 6A,B. Ordinary one-way ANOVA with Šidák's post hoc test was applied in Fig. 6A–C. Ordinary one-way ANOVA with Tukey's post hoc test was applied in Fig. 7A. Ordinary one-way ANOVA with Šidák's post hoc test was applied in Fig. 7B,D. Ordinary one-way ANOVA with Šidák's post hoc test was applied in Fig. 7E,F.

Ordinary one-way ANOVA with Tukey's post hoc test was applied in Fig. 7G. Unpaired $t$ test was applied in Fig. EV5B.

## Data availability

All raw RNA-seq data are available in the E-MTAB-14815 dataset (ENA database) at the https://www.ebi.ac.uk/biostudies/arrayexpress/studies/E-MTAB-14815?key=d4078a72-2b30-4035-b14a-5a5b4a95744d.

The source data of this paper are collected in the following database record: biostudies:S-SCDT-10_1038-S44318-025-00521-1.

## Peer review information

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

## Acknowledgements

We thank IFOM Cell Culture Facility for generation of cortical neurons from IPSCs. We thank Silvia Brambillasca of IFOM Experimental Therapeutic Unit for advice for Cell Viability assay. We thank IFOM Imaging facility for the support with imaging analyses. We thank the Mouse Genetics Service (Cogentech S.C.a.R.) the support with animal work. We thank Federica Pisati of the Histopathology Unit (Cogentech S.C.a.R.) for the support in preparation for brain tissue for all staining. We thank Claudia Balducci and Gianluigi Forloni from Department of Neuroscience of Mario Negri Institute for Pharmacological Research in Milan for providing advice on Aβ42 oligomers preparation. We

thank Professor Giorgio Binelli, Department of Biotechnology and Life Sciences University of Insubria, for the support in statistical analysis. We thank Costanzo Lab at IFOM for providing reagents. We thank Dr. Fabio Iannelli, IEO, for the preliminary analysis of RNAseq. F.d'A.d.F lab is supported by the following funds: ERC advanced grant TELORNAGING – 835103 (F.d'A.d.F.); ERC POC TELOVACCINE – 101113229 (F.d'A.d.F.); AIRC-IG 30471 (F.d'A.d.F.); AIRC-IG 21762 (F.d'A.d.F.); AIRC 5×1000 21091 (F.d'A.d.F.); Telethon GMR23T2007 (F.d'A.d.F.); Progetti di Ricerca di Interesse Nazionale (PRIN) 2020CXFL4T (F.d'A.d.F.); Progetti di Ricerca di Interesse Nazionale (PRIN) 2022R7LH5T (F.d'A.d.F.) AriSLA DDR&ALS FG_24_2020 (F.d'A.d.F.); POR FESR InterSLA DSB.AD004.294 (F.d'A.d.F.); Fondazione Regionale per la Ricerca Biomedica (Regione Lombardia) EJPRD19-206 PROGERIA, GA 825575 (F.d'A.d.F.); Next Generation EU, in the context of the National Recovery and Resilience Plan, Investment PE8 Project Age-It (F.d'A.d.F., F.C.); Investment CN3 National Center for Gene Therapy and Drugs based on RNA Technology (F.d'A.d.F.); #NEXTGENERATIONEU (NGEU) (F.C.).

## Author contributions

**Sara Sepe**: Conceptualization; Formal analysis; Investigation; Visualization; Methodology; Writing—original draft; Project administration. **Federica Rey**: Data curation; Formal analysis; Investigation; Visualization; Writing—review and editing. **Alexandra Mancheno-Ferris**: Data curation; Writing—review and editing. **Alessandra Bigi**: Investigation; Writing—review and editing. **Giulia Fani**: Investigation; Writing—review and editing. **Devid Damiani**: Investigation; Writing—review and editing. **Matteo Cabrini**: Investigation; Visualization; Writing—review and editing. **Eugenia Marinelli**: Investigation; Writing—review and editing. **Julio Aguado**: Investigation; Writing—review and editing. **Liliana Contu**: Investigation; Writing—review and editing. **Alessia di Lillo**: Investigation; Writing—review and editing. **Sara Boggio**: Investigation; Writing—review and editing. **Sara Tavella**: Investigation; Writing—review and editing. **Ilaria Rosso**: Investigation; Writing—review and editing. **Stefano Gustincich**: Supervision; Writing—review and editing. **Fabrizio Chiti**: Supervision; Writing—review and editing. **Fabrizio d'Adda di Fagagna**: Conceptualization; Supervision; Funding acquisition; Writing—original draft; Project administration.

Source data underlying figure panels in this paper may have individual authorship assigned. Where available, figure panel/source data authorship is listed in the following database record: biostudies:S-SCDT-10_1038-S44318-025-00521-1.

## Disclosure and competing interest statement

F.d'A.d.F. is an inventor on the patent applications RNA products and uses thereof (PCT/EP2013/ 059753) and therapeutic oligonucleotides (PCT/EP2016/068162). F.d'A.d.F. is a shareholder of TAG Therapeutics. The remaining authors declare no competing interests.

# Expanded View Figures

**Figure EV1.   Markers of tDDR activation and pro-apoptotic gene expression are elevated in the hippocampus of 3xTg-AD mice.**

(A–D) Hippocampal regions from 12-month-old wt and 3xTg-AD mice were analyzed. Each circle in the graphs represents one mouse analyzed. (A) Representative confocal images of immunofluorescence staining of hippocampal regions from wt and 3xTg-AD mice with antibodies against γH2AX, 53BP1, MAP2, nuclei were stained with DAPI. Bar graph shows quantification of the average number of γH2AX ($n = 4$ wt and $n = 4$ 3xTg-AD), and 53BP1 foci per nucleus ($n = 5$ wt and $n =$ 53xTg-AD). At least 50 nuclei per mouse section were scored. (B) Bar graphs show the quantification of G-rich ($n = 4$ wt and $n = 8$ 3xTg-AD) and C-rich ($n = 5$ wt and $n = 10$ 3xTg-AD) tdincRNA levels in 3xTg-AD mice relative to wt samples in brain hippocampus as measured by RT-qPCR. (C) Bar graph shows the quantification of p21 mRNA levels in 3xTg-AD mice relative to wt samples in the hippocampus as measured by RT-qPCR. ($n = 4$ wt and $n = 4$ 3xTg-AD). (D) Bar graph represents the quantification of the ratio between the mRNA levels of Bax over Bcl-2 in 3xTg-AD mice relative to wt samples in the hippocampus as measured by RT-qPCR ($n = 4$ wt and $n = 4$ 3xTg-AD). (E) Representative confocal microscopy images of Immunofluorescence stainings combining antibodies against MAP2, BAX and BCL-2 of sections from wt and 3xTg-AD mice (left panel). Bar graph represents the quantification of the ratio between the signal of BAX over BCL-2 from immunofluorescence in 3xTg-AD mice relative to wt samples in hippocampal regions ($n = 4$ wt and $n = 4$ 3xTg-AD). Unpaired $t$ test was applied and data are represented as mean ± SEM.

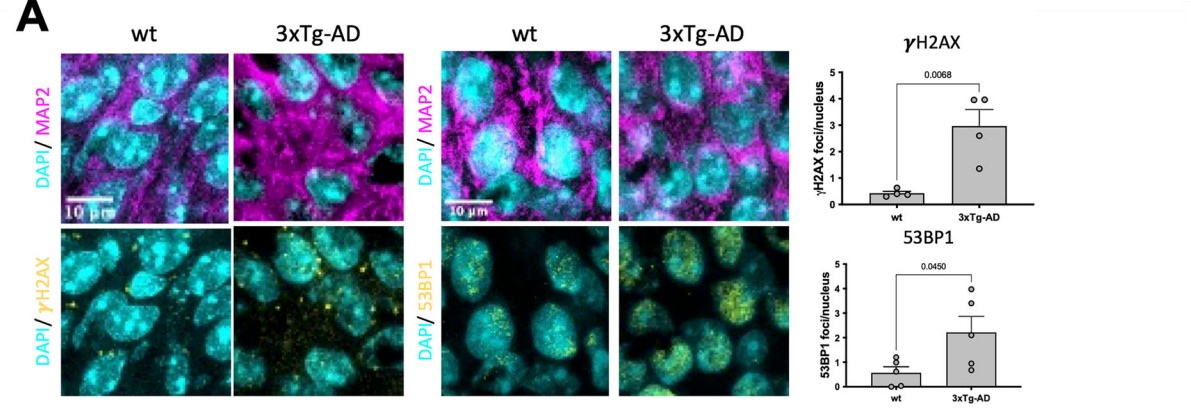

**A**

wt | 3xTg-AD | wt | 3xTg-AD

DAPI / MAP2 ... DAPI / γH2AX ... DAPI / MAP2 ... DAPI / 53BP1

γH2AX

53BP1

**B**

G-rich strand

C-rich strand

**C**                                          **D**

p21-mRNA levels                          Bax/Bcl-2 -mRNA

**E**

wt | 3xTg-AD

DAPI / MAP2

BAX / BCL2

BAX/BCL-2

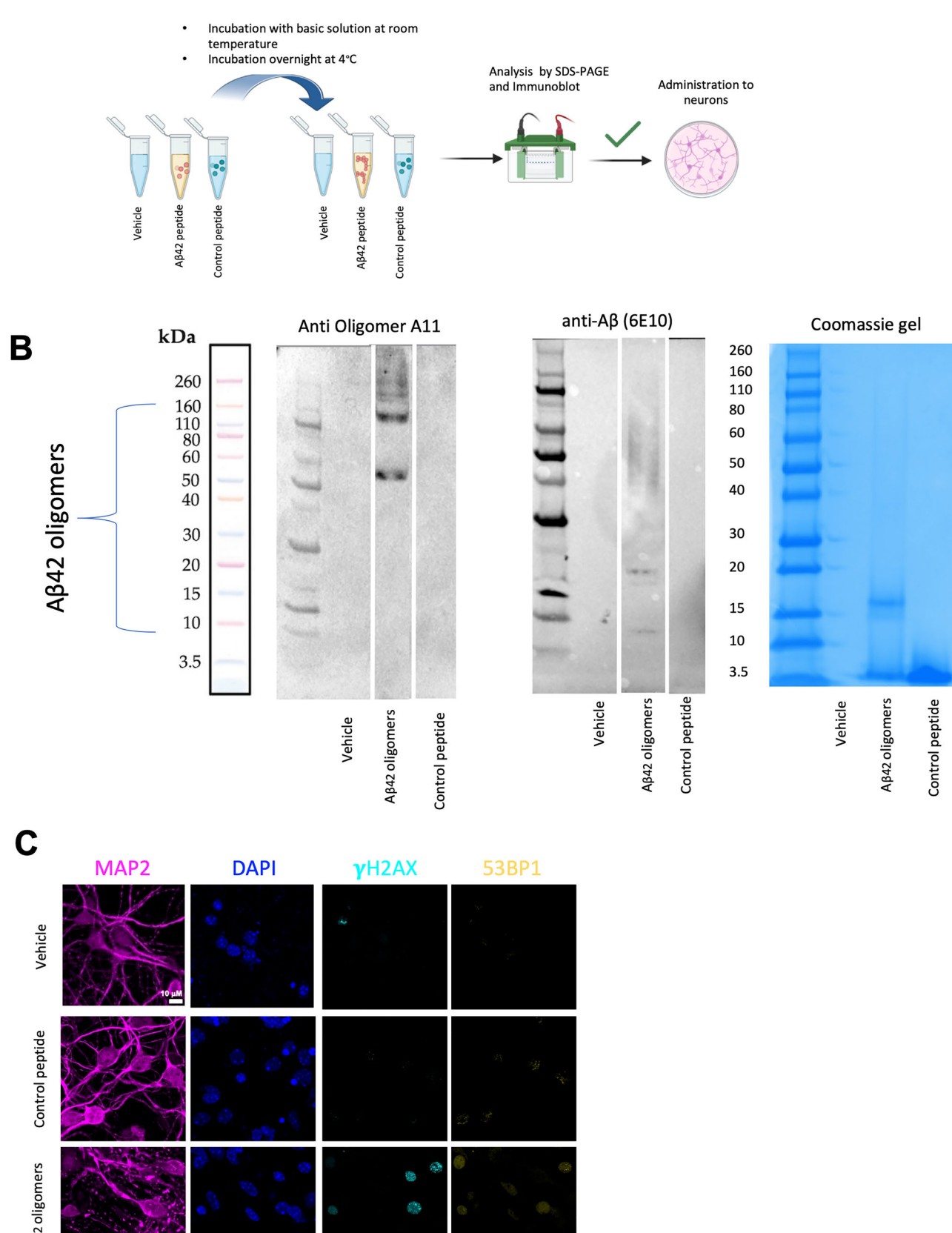

◄ **Figure EV2. Experimental descriptions of Aβ42 oligomers preparation and their specificity.**

In all experiments in this manuscript, a fresh batch of Aβ42 oligomers was prepared as described in the "Methods and protocols" section. (A) Scheme of Aβ42 oligomer preparation and analysis of the oligomer's solution. Each Aβ42 oligomers batch was tested by immunoblotting for oligomerization status before use. (B) Representative images of western blots using the Oligomers A11 polyclonal antibody recognizing selectively the oligomeric forms only, or the 6E10 monoclonal antibody recognizing the C terminal fragment (CTF) of the peptide. Control peptide prepared in parallel but unable to undergo oligomerization, was used as negative control. Polyacrylamide gel stained with blue Coomassie-based solution was used to check the Aβ42 oligomers and control peptide. (C) Representative images of immunofluorescence staining of mouse primary neurons treated with Vehicle, Aβ42 oligomers and solutions containing control peptide and stained with antibodies against MAP2 (magenta) as neuronal marker, γH2AX (cyan) and 53BP1 (yellow) as DDR markers, nuclei were stained with DAPI. Source data are available online for this figure.

**A**

|  | Vehicle | Aβ42 oligomers |
|---|---|---|
| DAPI | | |
| pS/TQ | | |
| pATM | | |

10 μM

**pATM**

0.0041

pATM foci/nucleus

Vehicle — Aβ42 oligomers

**pS/TQ**

0.0135

pS/TQ foci/nucleus

Vehicle — Aβ42 oligomers

**B**

|  | Vehicle | Aβ42 oligomers |
|---|---|---|
| DAPI | | |
| γH2AX | | |
| 53BP1 | | |

10 μM

**γH2AX**

0.0270

γH2AX foci/ nucleus

Vehicle — Aβ42 oligomers

**53BP1**

0.0008

53BP1 foci/nucleus

Vehicle — Aβ42 oligomers

**C**

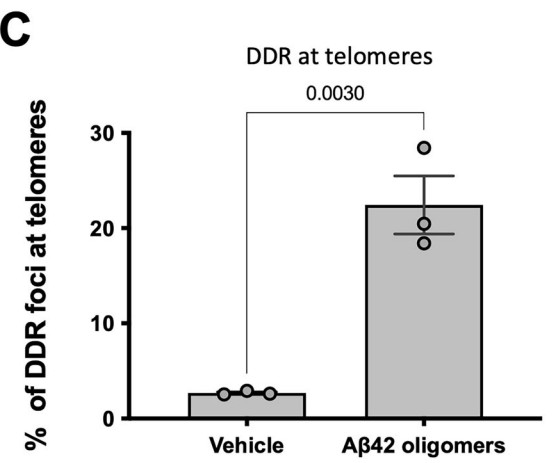

DDR at telomeres

0.0030

% of DDR foci at telomeres

Vehicle — Aβ42 oligomers

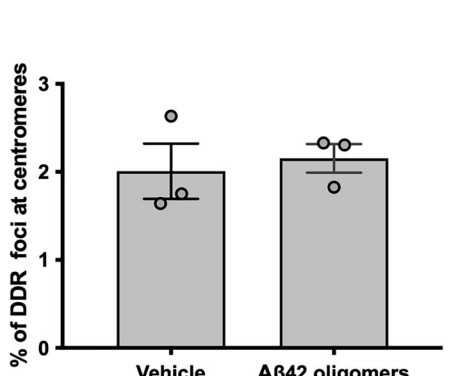

DDR at centromeres

% of DDR foci at centromeres

Vehicle — Aβ42 oligomers

◀ **Figure EV3.** **Aβ42 oligomers induce DDR and accumulation at telomeres in HT-22 cells.**

(A–C) HT-22 cells were treated with 5 μM Aβ42 oligomers for 24 h or vehicle. Each circle represents an independent biological replicate. (A) Representative confocal image of immunofluorescence staining of HT-22 with antibodies against pS/TQ (cyan) and pATM (yellow) as DDR markers, nuclei were stained with DAPI. Bar graphs show the quantification of average number of DDR foci per nucleus. At least 50 cells for each condition in each replicate were analyzed. ($n = 3$ for each condition) (B) Representative confocal images of immunofluorescence staining of HT-22 with antibodies against γH2AX (cyan) and 53BP1 (yellow) as DDR markers, nuclei were stained with DAPI. Bar graphs show the average number of DDR foci per nucleus. At least 50 cells for each condition in each replicate were analyzed. ($n = 3$ for each condition). (C) Bar graphs show the percentage of DDR foci that localize at telomeres or centromeres, as detected by immunoFISH combining γH2AX antibody and FISH for telomeric DNA using a complementary PNA probe and by double immunofluorescence combining antibodies against γH2AX and the centromeric protein CREST of primary neurons. At least 50 cells for each condition in each replicate were analyzed. ($n = 3$ for each condition). Unpaired $t$ test was applied and data are represented as mean ± SEM.

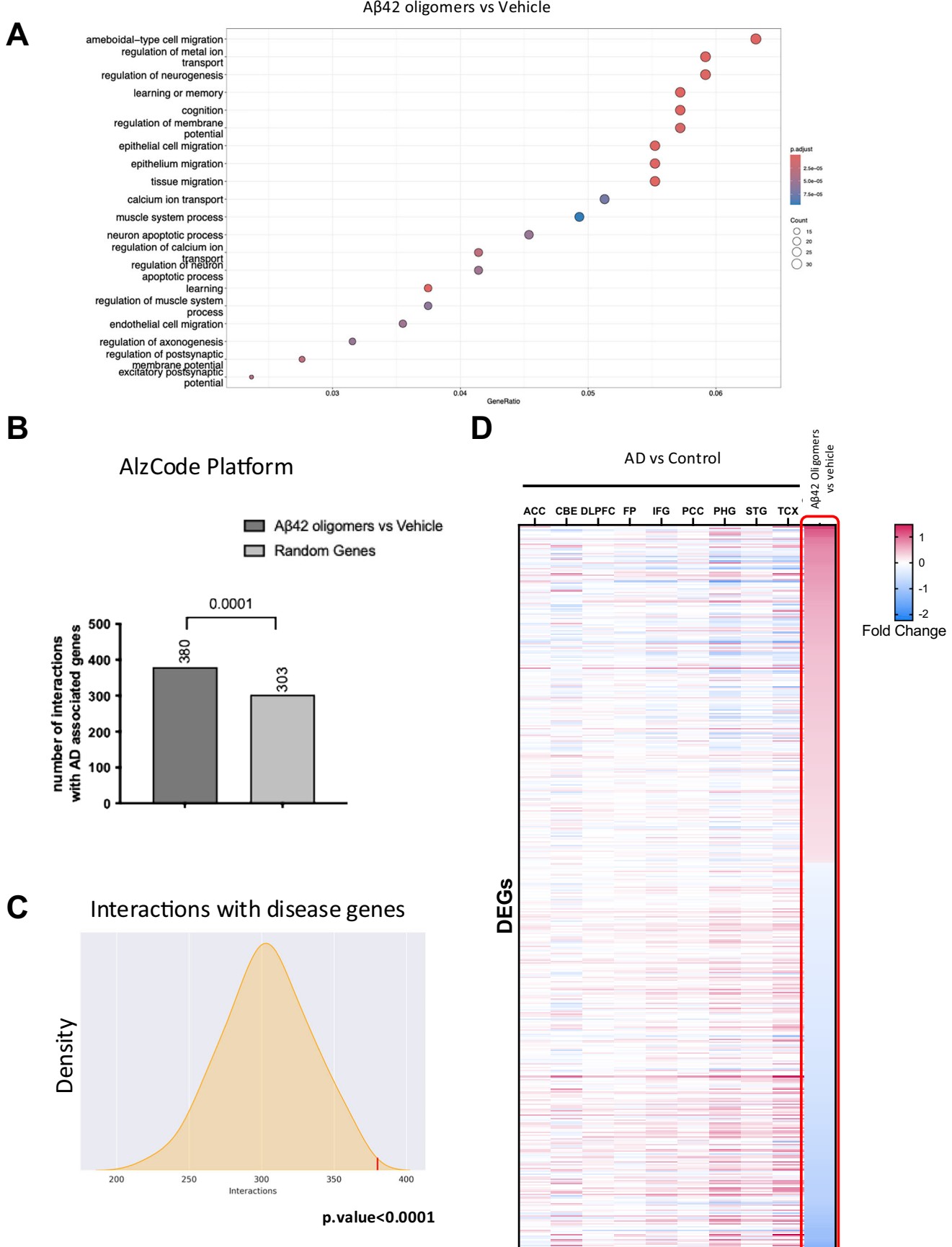

◀ **Figure EV4. Aβ42 oligomers induce an RNA-signature present in AD-affected areas.**

(A) The dotplot shows the significant Gene Ontology Biological processes that emerged from the enrichment analysis of DEGs in Aβ42 oligomers versus Vehicle, performed with ClusterProfiler. The name of the pathway is reported on the left of the graph, the *x*-axis represents the gene ratio, dots size represents the number of DEGs present in the pathway and the color indicates the adjusted *p* value of the pathway enrichment. Statistical analysis was performed with over-representation analysis for Gene Ontology Biological processes on DEGs, performed with ClusterProfiler (Wu et al, 2021). (B) The histogram shows the extent of correlation between DEG in Aβ42oligomers vs vehicle and known AD-associated genes when compared to random genes or known AD-associated genes. (C) Plot as generated by http://alzcode.xyz showing the number of the interactions of DEGs in Aβ42oligomers vs vehicle and known AD-associated genes (red line) compared to the distribution of interactions between random genes and known AD-associated genes. The significance is calculated by permutation test (Lin et al, 2022). (D) Heatmap showing the 477 DEGs in Aβ42 oligomers-treated vs vehicle-treated primary neurons and in human AD patients vs control patients in the indicated brain areas. ACC anterior cingulate cortex, CBE cerebellum, DLPFC dorsolateral prefrontal cortex, FP frontal pole, IFG inferior frontal gyrus, PCC posterior cingulate cortex, PHG parahippocaml gyrus, STG superior temporal gyrus, TCX temporal cortex.

**A**

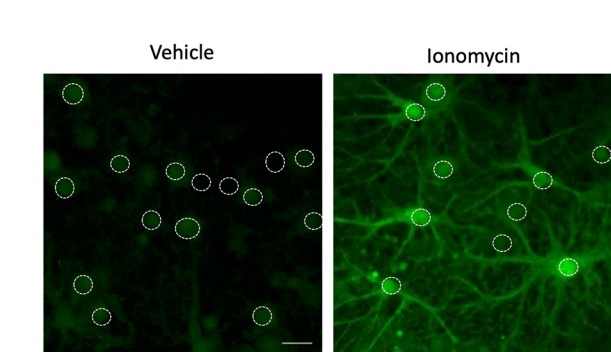

Vehicle          Ionomycin

**B**

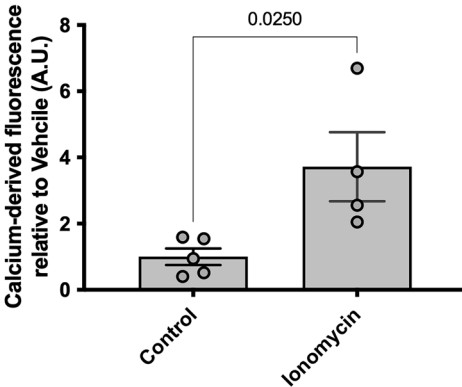

**Figure EV5.   Positive control for Ca$^{2+}$ measurements in mouse primary neurons.**

Treatment of mouse primary neurons with 2 μM ionomycin for 2 h was used as positive control for Ca$^{2+}$-derived fluorescence signal. Each circle represents an independent biological replicate. (**A**) Representative confocal microscopy images of mouse primary neurons incubated with Fluo-4 probe, green fluorescence is generated upon Ca$^{2+}$ binding to the Fluo-4 probe. White dashed-line circles indicate nuclei of neurons. White dashed-line circles indicate the nuclei of neurons. Bar correspond to 20 μm. (**B**) Bar graph shows semiquantitative measures of intracellular Ca$^{2+}$-derived fluorescence. (control, $n = 5$; Ionomycin, $n = 4$) Unpaired $t$ test was applied and data are represented as mean ± SEM.

## A

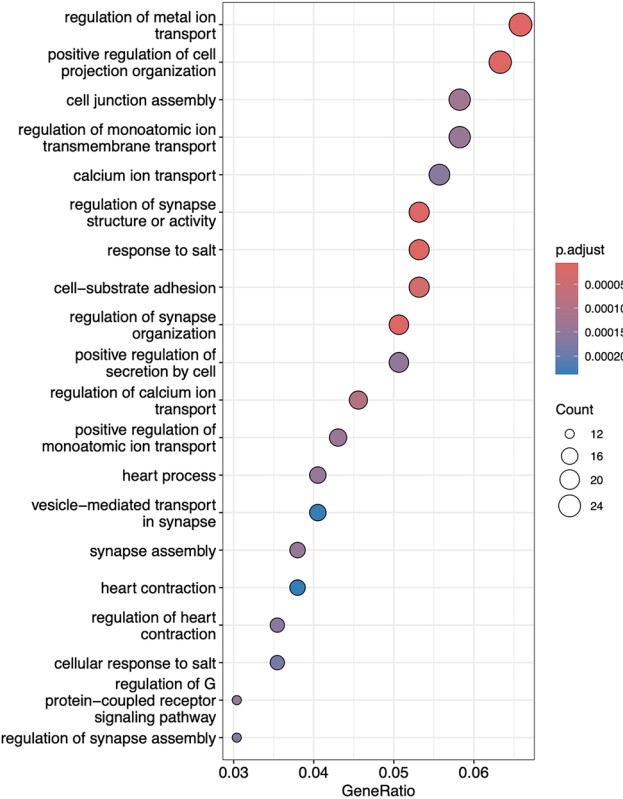

### GO Biological Processes
Aβ42 oligomers Anti −teloG vs Aβ42 oligomers Control ASO

**Figure EV6. tASO improves pathways dysregulated by Aβ42 oligomers.**

(**A**) The dotplot shows the significant Gene Ontology Biological processes that emerged from the enrichment analysis of DEGs in Aβ42 oligomers Anti-Telo G-treated vs Aβ42 oligomers Control ASO-treated primary neuron, performed with ClusterProfiler. The name of the pathway is reported on the left of the graph, the *x*-axis represents the gene ratio, the dot size represents the number of differentially expressed genes present in the pathway and the color indicates the adjusted *p* value of the pathway enrichment. Statistical analysis was performed with over-representation analysis for Gene Ontology Biological processes on DEGs, performed with ClusterProfiler (Wu et al, 2021).

