## [Peer Review File · The EMBO Journal]

Telomeric DNA damage response mediates neurotoxicity of A β 42 oligomers in Alzheimer's disease

Sara Sepe, Federica Rey, Alexandra Mancheno-Ferris, Alessandra Bigi, Giulia Fani, Devid Damiani, Matteo Cabrini, Eugenia Marinelli, Julio Aguado, Liliana Contu, di Lillo Alessia, Sara Boggio, Sara Tavella, Ilaria Rosso, Stefano Gustincich, Fabrizio Chiti, and Fabrizio d'Adda di Fagagna

Corresponding author(s): Fabrizio d'Adda di Fagagna (fabrizio.dadda@igm.cnr.it)

Review Timeline:

Submission Date:	18th Nov 24
Editorial Decision:	18th Dec 24
Revision Received:	7th Apr 25
Editorial Decision:	12th Jun 25
Revision Received:	1st Jul 25
Accepted:	4th Jul 25

Editor: Ioannis Papaioannou

Transaction Report:

Dear Dr. d'Adda di Fagagna,

Thank you again for submitting your manuscript EMBOJ-2024-119642 for consideration by The EMBO Journal, and for your patience during peer review. Your manuscript has now been seen by three experts in the field, and we have received the full set of their well-informed and detailed reports, which are included below.

As you will see, the referees recognize that the amount of the work presented in this manuscript is commendable, the results are interesting, and the overall advance to the field significant. However, they also identify a number of limitations in the study and the manuscript, and they provide reasonable suggestions for strengthening the experimental work, the analyses, and the presentation of the results in the manuscript further. We agree with the referees that the manuscript and its impact on the field would significantly benefit from the suggested improvements.

Given the referees' positive comments and recommendations, I would like to invite you to submit a revised version of the manuscript along with a detailed point-by-point response addressing all referees' comments. I should add that it is The EMBO Journal policy to allow only a single round of major revision, and acceptance of your manuscript will therefore depend on the completeness of your responses in this revised version. Please let me know if you have any questions or comments that you would like to discuss with me.

We generally allow three months as standard revision time (March 17, 2025). As a matter of policy, competing manuscripts published during this period will not negatively impact our assessment of the conceptual advance presented by your study. However, we request that you contact us as soon as possible upon publication of any related work, to discuss how to proceed. Should you foresee a problem in meeting this three-month deadline, please let us know in advance and we may be able to grant an extension.

Thank you for the opportunity to consider your work for publication in The EMBO Journal. I look forward to your revision.

Best regards,

Ioannis

Instructions for preparing your revised manuscript

1. When you are ready to submit the revision, please upload:

- A Word file of the manuscript text (including legends of main Figures, EV Figures and Tables). Please make sure that changes are highlighted (or "tracked") to be clearly visible.

- Individual production-quality figure files (one file per figure). When assembling your figures, please refer to our figure preparation guidelines in order to ensure proper formatting and readability in print as well as on screen:

If the data shown in a figure are obtained from n {less than or equal to} 2, please use scatter plots showing the individual data points.

- i. the name of the statistical test used to generate error bars and P values
- ii. the number (n) of independent experiments (please specify technical or biological replicates) underlying each data point (discussion of statistical methodology can be reported in the Materials and Methods section, but figure legends should contain a basic description of n , P , and the test applied)
- iii. the nature of the bars and error bars (s.d., s.e.m.).

- A point-by-point response to the referees' comments, with a detailed description of the changes made (as a word file). All referees' concerns must be fully addressed and their suggestions taken on board. When preparing your letter of response to the

referees' comments, please bear in mind that this will form part of the Review Process File and will therefore be available online to the community. Please note that you have the possibility to opt out of the transparent process at any stage prior to publication by letting the editorial office know (contact@embojournal.org); if you do opt out, the Review Process File link will point to the following statement: "No Review Process File is available with this article, as the authors have chosen not to make the review process public in this case.". For more details on our Transparent Editorial Process, please visit our website: <https://www.embopress.org/page/journal/14602075/authorguide#transparentprocess>

- Expanded View (EV) files (replacing Supplementary Information) that are collapsible/expandable online. A maximum of 5 EV Figures can be typeset. EV Figures should be cited as "Figure EV1, Figure EV2" etc. in the text, and their respective legends should be included in the manuscript file after the legends of regular figures. See detailed instructions regarding Expanded View files here:

- For the figures that you do NOT wish to display as Expanded View figures, they should be bundled together with their legends in a single PDF file called "Appendix", which should start with a short Table of Contents (including page numbers). Appendix figures should be referred to in the main text as: "Appendix Figure S1, Appendix Figure S2" etc. Please see detailed instructions here: <https://www.embopress.org/page/journal/14602075/authorguide#expandedview>

- A complete author checklist, which you can download from our author guidelines (<https://www.embopress.org/page/journal/14602075/authorguide>). Please note that the checklist will also be part of the Review Process File.

2. Please note that no statistics should be calculated and shown in Figures if $n=2$. Please also note that each p value should be reported as an exact value.

3. Before submitting your revision, primary datasets (and computer code, where appropriate) produced in this study need to be deposited in appropriate public databases (see <https://www.embopress.org/page/journal/14602075/authorguide#dataavailability>).

In particular, we kindly request you to deposit the RNA sequencing data produced in your study to an appropriate database. The accession numbers, database, and the specific URLs (links) should be listed in a formal "Data availability" section (placed after Methods), following the example below:

"The RNA-seq datasets produced in this study are available in the following database:
Gene Expression Omnibus GSE46843 (<https://www.ncbi.nlm.nih.gov/geo/query/acc.cgi?acc=GSE46843>)"

*** All links should resolve to a page where the data can be accessed. ***

*** Please remember to provide in the Data availability section of your revised manuscript reviewer passwords if the datasets are not yet public. ***

*** The Data Availability Section is restricted to new primary data that are part of this study. In case you have no data that require deposition in a public database, please state so instead of referring to the database: "Our study includes no data deposited in public repositories." under the heading "Data availability". ***

4. Please check that the title and the abstract of the manuscript are brief, yet explicit, even to non-specialists. The length of the title should not exceed 100 characters, and the abstract should be a single paragraph not exceeding 175 words.

5. Please also note our reference format: <https://www.embopress.org/page/journal/14602075/authorguide#referencesformat>.

7. Please remember: digital image enhancement is acceptable practice, as long as it accurately represents the original data and conforms to community standards. If a figure has been subjected to significant electronic manipulation, this must be noted in the figure legend or in the "Materials and Methods" section. The editors reserve the right to request original versions of figures and the original images that were used to assemble the figure.

8. Our journal encourages inclusion of data citations in the reference list to directly cite datasets that were obtained from public databases. Data citations in the article text are distinct from normal bibliographical citations and should directly link to the database records from which the data can be accessed. In the main text, data citations are formatted as follows: "Data ref: Smith et al, 2001" or "Data ref: NCBI Sequence Read Archive PRJNA342805, 2017". In the Reference list, data citations must be labeled with "[DATASET]". A data reference must provide the database name, accession number/identifiers, and a resolvable link to the landing page from which the data can be accessed at the end of the reference. Further instructions are available at:

<https://www.embopress.org/page/journal/14602075/authorguide#referencesformat>.

9. We request authors to consider both actual and perceived competing interests. Please review our policy (<https://www.embopress.org/page/journal/14602075/authorguide#conflictsofinterest>) and update your competing interests statement if necessary. Please name this section 'Disclosure and competing interests statement' and place it after the Acknowledgements section.

10. Please note that all corresponding authors are required to provide an ORCID ID upon submission of a revised manuscript (<https://orcid.org/>). Please find instructions on how to link your ORCID ID to your account in our manuscript tracking system in our Author guidelines (<https://www.embopress.org/page/journal/14602075/authorguide#authorshipguidelines>).

11. We use CRediT to specify the contributions of each author in the journal submission system. CRediT replaces the author contribution section, which should be removed from the manuscript. Please use the free text box to provide more detailed descriptions. See also guide to authors: <https://www.embopress.org/page/journal/14602075/authorguide#authorshipguidelines>.

13. We would also welcome the submission of cover suggestions or motifs to be used by our Graphics Illustrator in designing a cover.

14. Please use the link below to submit your revision:

XXXXXXXXXXXXXXXXXX

Referee #1:

The study investigates the role of tDDR in AD, focusing on the neurotoxic effects of A β 42 oligomers. It shows that tDDR, triggered by ROS and calcium imbalance, contributes to neuronal damage and AD pathology. By using ASOs to inhibit tDDR, the study reduces neurotoxicity in both mouse and human neurons, offering a potential new therapeutic approach for AD. While oxidative stress and its role in A β toxicity are well known, this study provides new insight into how oxidative stress leads to telomeric DNA damage and RNA-based mechanisms that drive neurodegeneration.

Overall, the study is well structured, and its conclusions are mostly supported by the experimental work. However, the following observations could further strengthen the study:

Although the presence of elevated DDR markers and tdincRNAs is shown, the direct functional link between these markers and cognitive deficits in the 3xTg-AD model is not validated and should be acknowledged or further explored.

The study focuses on a single age (12 months), missing potential insights into when telomeric damage begins in the disease course. Is it progressive?

In certain instances, a more extensive cohort (>3 mice) could help account for variability.

The use of 10 μ M A β 42 oligomers for 48 hours in vitro might not reflect physiological concentrations in vivo. In vivo or dose-response studies are not included.

While the study establishes causation between A β 42 oligomers and tDDR, it does not show whether blocking tDDR (e.g., using antioxidants or tASOs) prevents neuronal damage in this specific experiment.

Analysis of the RNA-seq data for known DNA repair genes could clarify whether DNA repair pathways are directly disrupted by A β 42 oligomers.

While the study demonstrates short-term efficacy, long-term treatment with tASOs in human neurons or animal models has not been assessed. Chronic use of tASOs might have unforeseen consequences, especially in the context of telomere biology. If there is existing knowledge on this, it should be commented.

Referee #2:

EMBO J-2024-119642

Sara Sepe et al. showed that the cortex and hippocampus of 3xTg-AD mice accumulate markers of DNA Damage Response (DDR) at telomeres. They also showed that mouse primary hippocampal neurons and iPSC-derived human cortical neurons exposed to 42-residue A β oligomers trigger telomeric DDR, which is dependent on increased levels of reactive oxygen species

caused by calcium imbalance. Furthermore, the authors demonstrated that inhibition of A β 42-induced telomeric DDR by antisense oligonucleotides (ASOs) targeting RNA generated at damaged telomeres reduces neurotoxicity in mouse and human neurons and restores transcriptional pathways altered by A β .

Based on these results the authors suggested a novel opportunity for RNA-based therapies for Alzheimer's disease (AD).

This paper provides very interesting scope of AD with a novel opportunity for RNA-based therapies, and experiments were well planned and performed.

The reviewer has some comments to improve this manuscript before publication.

Major comments

1. In Figure 1A, B: It is not clear whether the nuclei shown in the figure are neuronal nuclei or not. Please provide counter-staining with some neuronal marker such as NeuN.
2. Results, page 5, lines 14-16: The authors described "Similar to the observation in vivo, we observed an increased levels of p21 mRNA in mouse neurons treated with A 42 oligomers compared with vehicle (Fig. 2F)." This is not true because there is no significant increase of p21 mRNA level in 3xTg-AD hippocampus as shown in Expanded View Fig.1 C.
3. Results, page 5, lines: "The analysis of the fraction of DDR-positive telomeres or centromeres shows that the telomeres of neurons treated with A 42 oligomers are preferentially damaged when compared with vehicle-treated neurons (Fig.2D)." Results, page 5, lines 34-36: "In summary, we demonstrated that A β 42 oligomers induce DDR activation in mouse hippocampal primary neurons, and that telomeres are preferential sites of DDR recognition as independently determined by colocalization studies and tdnRNA levels." As seen in Fig.2C, DDR at telomeres is about 20%, and that in centromeres are less than 4 %, that means majority of DDR exists in non-telomeric/non-centromeric region of chromosomes. "telomeres are preferential sites of DDR recognition" is not appropriate to conclude. Please edit appropriately.
4. Fig.2B: It is not clear what the arrows in Figure 2B are pointing to.
5. Fig.2D: Numbers of Y axis is too small compared to Fig.2C or Fig1D.
6. Fig. 2I, Fig.6F: please provide more detailed explanation for p-value, Genes, and GeneRatio in the legend.
7. Please provide more information for the CM-H2DCFDA, especially types of ROS which can be detected.
8. Results page 6, lines 28-29: "These results demonstrate that DDR and more specifically tDDR activation, is caused by ROS generated upon exposure of mouse primary neurons to A 42 oligomers." In Fig. 3D, E, only 10% of DDR is at telomeres, and more than 50% of DDR was suppressed by NAC, so ROS cause general DDR not specifically tDDR.
9. Fig. 5B: Please show % of DDR-positive telomeres, and also ROS level should be examined to show that tASOs do not alter ROS production.

Referee #3:

Authors: Sepe et al.

Title: Telomeric DNA damage response activation mediates A β 42 oligomers neurotoxicity in Alzheimer disease

The manuscript presents data showing that in the cortex and hippocampus of 3xTg-AD mice there is an activation of the DNA damage response, specifically at telomeres. The work expands these observations to include iPSC-derived human cortical neurons. Exposure to A β 42 enhances this response, purportedly due to increased levels of reactive oxygen species. The authors further maintain that this damage is caused by an imbalance in calcium homeostasis. Then, using antisense oligonucleotides against RNA generated at telomeres reduced neuronal death. The authors then proposed that the data point to the potential of RNA-based therapies for Alzheimer's disease.

The Introductory section captures many of the relevant topics and references, but as written it is choppy and would benefit from some editing to smooth the flow of logic. One problem the authors face is that while many of the phenomena described are correlated with one another, the arrows of causality have not been rigorously shown. This is most evident in the linkages between A β , ROS, NMDA-R, Ca²⁺, and DNA damage. Attention to this detail would improve readability

Figure 1 is described as being from "brain cortex". This is not adequate given that DNA damage is likely variable from one region to the next and at different cortical depths. The 53BP1 staining image (Fig. 1A) would be better if red and green were used to show overlap of nuclear foci instead of yellow and cyan (also the meaning of the arrows in the figure are not explained in the legend). The use of γ -H2AX as a DNA damage marker is appropriate but lacks precision. The phosphorylation of H2AX spreads from the break as repair proceeds and with time is spatially less accurate. One possibility to consider is that the modified histone persists at the telomere even after the damage itself has been repaired. A direct assay of a DNA break such as the TUNEL reaction would be an important validation of the claim that damage persists at the telomere.

The use of p21 as a damage marker is inadequate by itself as it is also a marker of senescence and direct cell cycle activity inhibition. qPCR of message levels loses all spatial and cell-type information. This is important because p21 is elevated in neurons in AD brains with no signs of punctate staining as would be expected if telomeres were labeled (see references from Thomas Arendt). Efforts should be made to assay other more direct indices downstream of the DDR - components of the MRN complex for double strand breaks, PARP1 and others for single strand breaks.

The Bcl2/BAX ratio of qPCR signals is a very imprecise way of addressing the question of cell death. Even with the dissection to enrich for granule cells there will be many other cell types present, and the source of the signal could be coming from anywhere. Immunocytochemistry would be preferable in this situation.

The nature of the A β preparation is worrisome. The western blots in EVFig. 2 show no obvious difference between the 6E10 and A11 stain. The presence of a monomeric band in the A11 lane is especially troubling as the antibody purportedly does not label the non-aggregated forms of the peptide. Further, the application of A β increases γ -H2AX staining, but close inspection of Fig. 2B reveals that the location is adjacent to, not overlapping with, the telomeric labeling. This tends to undercut the authors' argument and should be clarified. Again, the meaning of the white arrows is not explained and is not obvious from their location.

The RNAseq data is confusing. The X-axis of the volcano plot in Fig. 2G is compressed, presumably to include the four outlier points, but the impression is that with the exception of these four, there is very little change in gene expression. The high significance is noted, but the finding that gene expression is little changed is surprising given that application of oligomers is such a strong stimulant. The genes that are most changed or most significant should be identified. Consulting the Agora data makes sense, but the brain regions identified are not revealed, nor are they validated in the mouse brain. This diminishes the power of the analysis. Fig. EV6B is also a confusing presentation of the data. The final two columns are consistent with the authors' claims about the effect of the ASO, but controls for the effects of ASO alone are missing (no A β). The rationale for showing the brain region data is not clear as there is no clear relationship between the two right hand columns and the rest.

Overall, a considerable amount of work has gone into this study, but the presentation and the analysis are flawed in important ways and the strength of the support for the authors' hypotheses is therefore diminished.

Minor points:

When discussing replicates, the authors should clarify whether these are technical or biological replicates (multiple coverslips from one culture or multiple, independent, cultures)

The abstract mentions human iPSC neurons, but nearly all of the work is done with mouse cells. A restructuring of the text would be advised.

The involvement of cell death should be monitored in more experiments. The DAPI images (e.g., Fig. 4C suggest that the oligomers induced a fair amount of apoptosis. If correct, this would alter the interpretation of the findings.

Referee #1:

The study investigates the role of tDDR in AD, focusing on the neurotoxic effects of A β 42 oligomers. It shows that tDDR, triggered by ROS and calcium imbalance, contributes to neuronal damage and AD pathology. By using ASOs to inhibit tDDR, the study reduces neurotoxicity in both mouse and human neurons, offering a potential new therapeutic approach for AD. While oxidative stress and its role in A β toxicity are well known, this study provides new insight into how oxidative stress leads to telomeric DNA damage and RNA-based mechanisms that drive neurodegeneration.

Overall, the study is well structured, and its conclusions are mostly supported by the experimental work. However, the following observations could further strengthen the study:

Although the presence of elevated DDR markers and tdincRNAs is shown, the direct functional link between these markers and cognitive deficits in the 3xTg-AD model is not validated and should be acknowledged or further explored.

We appreciate the referee's comment. To demonstrate a direct causal link between elevated tDDR markers, tdincRNAs and cognitive deficits in this AD model, we are presently setting up conditions to explore the impact of tDDR inhibition by tASO *in vivo* by intraventricular injections. We plan to disseminate these results in a follow up publication. Meanwhile, we have explicitly acknowledged this limitation in the revised manuscript in page 9, lines 24-26.

The study focuses on a single age (12 months), missing potential insights into when telomeric damage begins in the disease course. Is it progressive?

Stimulated by this referee, we performed additional analyses of the same cohort of mice at 6 months. At this age, we did not observe significant differential accumulation of the DDR marker γ H2AX in 3XTg-AD mice compared to age-matched wt mice – see below.

These results are consistent with a study showing severe cognitive decline starting at 12 months of age (Belfiore et al, *Aging cell*, 2019)

In certain instances, a more extensive cohort (>3 mice) could help account for variability.

We have now included samples from additional mice to increase the number of hippocampi analysed for DDR markers and apoptotic genes and now all analyses include from 4 up to 9 animals.

The use of 10 μ M A β 42 oligomers for 48 hours *in vitro* might not reflect physiological concentrations *in vivo*. *In vivo* or dose-response studies are not included.

While we agree that the concentrations used *in vitro* may not mirror concentrations *in vivo*, they are fairly standard in the field (i.e. Suberbielle et al, Nature Neuroscience, 2013; Hung et al Scientific Reports, 2015). This is for the purpose of modelling events occurring over a long period of time in an experimentally-manageable setup.

The observations, reported in our manuscript and in the previous reports cited, that this *in vitro* concentration recapitulates events occurring *in vivo*, suggest that they reproduce, in an accelerated manner, pathologically relevant events.

While the study establishes causation between A β 42 oligomers and tDDR, it does not show whether blocking tDDR (e.g., using antioxidants or tASOs) prevents neuronal damage in this specific experiment.

We would like to point out that we did show that blocking tDDR by tASO reverts neurotoxicity induced by A β 42 oligomers. Indeed, tASO treatments rescue the DNA damage response (Fig.5A-D, Fig 6A, Fig.7A-E), neuronal viability (Fig.6B, and Fig.7F), and apoptotic genes expression (Fig. 6C and Fig. 7G). Conclusions were consistently drawn in in both mouse and human neurons.

In addition, in mouse primary neurons we showed that the tASO treatment reverts transcriptional pathways induced by A β 42 oligomers, shared with AD patients (Fig 6 D-G).

Regarding the the use of antioxidants, we explored the impact of NAC on reactive oxygen species (ROS) and DNA damage response (Fig 3 A-B and D-F). Now, stimulated by this referee, we performed a cell viability assay upon NAC treatment and observed a beneficial effect on mouse primary neurons, and we are excited to include in the revised manuscript this result (Fig. 3C, page 6 lines 28-30)

These conclusions align well with the literature, which highlights the beneficial effects of antioxidant-based interventions in Alzheimer's disease, particularly in the context of A β amyloid pathology (Guo et al, 2025, Ageing Research Reviews).

Analysis of the RNA-seq data for known DNA repair genes could clarify whether DNA repair pathways are directly disrupted by A β 42 oligomers.

We thank the referee for the comment. We have performed additional analyses of our RNA-Seq dataset and further inspection of the Gene Ontology biological process did not show significant changes in DNA repair pathway (GO:0006281). We have now mentioned this result in the revised manuscript (page 6, lines 1-4). In addition, stimulated by the referee, we performed an additionally bioinformatic analysis to search for significantly deregulated genes within the DNA Repair pathway Hallmark. We identified only 6 genes which however are not implicated in the canonical DNA repair pathways but are rather secondary players in the process, as summarized in the table below for this referee.

Gene	Encoded Protein	Function	Association with DNA Repair
Srsf6	Serine/arginine-rich splicing factor 6 (SRSF6)	An RNA-binding protein involved in alternative splicing.	SRSF6 regulates alternative splicing of genes enriched in DDR and DNA repair pathways (Yang et al, 2020 Oncol Rep.)
Arl6ip1	ADP-ribosylation factor-like 6 interacting protein 1 (ARL6IP1)	Involved in membrane trafficking	No strong direct link
Npr2	Natriuretic peptide receptor 2 (NPR2)	A receptor that binds natriuretic peptides, playing a role in cardiovascular homeostasis.	No strong direct link
Pde4b	Phosphodiesterase 4B (PDE4B)	Enzyme that hydrolyzes cAMP, involved in signal transduction.	No strong direct link
Aaas	Aladin WD repeat nucleoporin (ALADIN)	Component of the nuclear pore complex, involved in nucleocytoplasmic transport.	Mutations in AAAS are linked to triple-A syndrome, and alter import of DNA Repair proteins (Aprataxin and DNA Ligase I) (Hirano et al., PNAS 2006)
Bcam	Basal cell adhesion molecule (BCAM)	Cell surface glycoprotein involved in cell adhesion.	No strong direct link.

In addition, to functionally validate these computational analyses and experimentally strengthen our conclusions, we performed a DNA repair assay in which HT-22 cells, immortalized neuronal cells, were treated with A β 42 oligomers for 24h and exogenously damaged by exposing them to ionizing radiations (5Gy). DNA damage generation and repair, as measured by the quantification of the number of γ H2AX foci per cell revealed that A β 42 oligomer-treated neuronal cells do not exhibit delayed or impaired DNA damage repair (see graphs below), indicating that A β 42 oligomers do not impair endogenous DNA repair activities.

While the study demonstrates short-term efficacy, long-term treatment with tASOs in human neurons or animal models has not been assessed. Chronic use of tASOs might have unforeseen consequences, especially in the context of telomere biology. If there is existing knowledge on this, it should be commented.

These data were removed from the point-by-point response as confidential, but they were provided by the authors during peer review for the information of the editorial team and the reviewers.

Referee #2:

EMBO J-2024-119642

Sara Sepe et al. showed that the cortex and hippocampus of 3xTg-AD mice accumulate markers of DNA Damage Response (DDR) at telomeres. They also showed that mouse primary hippocampal neurons and iPSC-derived human cortical neurons exposed to 42-residue A β oligomers trigger telomeric DDR, which is dependent on increased levels of reactive oxygen species caused by calcium imbalance. Furthermore, the authors demonstrated

that inhibition of A β 42-induced telomeric DDR by antisense oligonucleotides (ASOs) targeting RNA generated at damaged telomeres reduces neurotoxicity in mouse and human neurons and restores transcriptional pathways altered by A β . Based on these results the authors suggested a novel opportunity for RNA-based therapies for Alzheimer's disease (AD).

This paper provides very interesting scope of AD with a novel opportunity for RNA-based therapies, and experiments were well planned and performed. The reviewer has some comments to improve this manuscript before publication.

Major comments

1. In Figure 1A, B: It is not clear whether the nuclei shown in the figure are neuronal nuclei or not. Please provide counter-staining with some neuronal marker such as NeuN.

As suggested by the reviewer, we performed several tests using several antibodies against NeuN. However, we were unable to find an antibody that works reliably in combination with our DDR markers. Given this limitation, we used MAP2, a well-established neuronal marker that we consistently use throughout the paper to identify neurons, as a valid alternative to address the reviewer's request. Thus, in the revised manuscript, we incorporated images in Figure 1A/1B and EV 1A showing co-staining with MAP2 which demonstrates that neurons are positive for DDR and tDDR markers.

2. Results, page 5, lines 14-16: The authors described "Similar to the observation in vivo, we observed an increased levels of p21 mRNA in mouse neurons treated with A β 42 oligomers compared with vehicle (Fig. 2F)." This is not true because there is no significant increase of p21 mRNA level in 3xTg-AD hippocampus as shown in Expanded View Fig.1 C.

The referee's point is valid, since a significant increase of p21 was detected in the isocortex but not in the hippocampal region. Based on the referee's comment, we have rephrased the sentence to clarify this (page 5 lines 16-18).

3. Results, page 5, lines: "The analysis of the fraction of DDR-positive telomeres or centromeres shows that the telomeres of neurons treated with A β 42 oligomers are preferentially damaged when compared with vehicle-treated neurons (Fig.2D)." Results, page 5, lines 34-36: "In summary, we demonstrated that A β 42 oligomers induce DDR activation in mouse hippocampal primary neurons, and that telomeres are preferential sites of DDR recognition as independently determined by colocalization studies and tddincRNA levels." As seen in Fig.2C, DDR at telomeres is about 20%, and that in centromeres are less than 4 %, that means majority of DDR exists in non-telomeric/non-centromeric region of chromosomes. "Telomeres are preferential sites of DDR recognition" is not appropriate to conclude. Please edit appropriately.

Thank you for this thoughtful feedback regarding the interpretation of the data presented in Fig. 2C and the statement on page 5. This is for us an occasion to clarify our point. Telomeres represent only about 0.1% of the total genome, thus any random co-localization should then be

extremely rare. Therefore, our observation that 15-20% of DDR markers accumulate at telomeres supports the notion that telomeres are preferentially damaged by A β 42 oligomers. To additionally support our conclusions with a quantitative analysis, after angular transformation of the data, a two-way ANOVA was performed considering both experimental conditions (Vehicle and A β 42 oligomers) and the location of DDR markers (centromere, telomere and not centromeric/not telomeric DNA). The analysis of the variance showed that the amount of damage is significantly more pronounced at the telomeres than at the centromeres, as showed by the graph below which we show here for the referee.

This result confirms that telomere damage is significantly higher than what would be expected from a random distribution, supporting the idea that, relative to their abundance, telomeres are preferentially damaged by A β 42 oligomers. We have now made this point clear in the revised manuscript (page 5 lines 30-33).

4. Fig.2B: It is not clear what the arrows in Figure 2B are pointing to.

We apologize for the mistake. In the revised manuscript we adjusted the arrows pointing correctly to colocalization events.

5. Fig.2D: Numbers of Y axis is too small compared to Fig.2C or Fig1D.

In the revised manuscript we have modified the size of the numbers of the Y axis in Fig. 2D.

6. Fig. 2I, Fig.6F: please provide more detailed explanation for p-value, Genes, and GeneRatio in the legend.

We have edited for clarity figure legends of Fig. 2I and Fig. 6F as requested. In Fig. 2I: “The dotplot shows the 20 most significant Gene Ontology Biological processes that emerged from the enrichment analysis of “concordant DEGs” performed with ClusterProfiler. The name of the pathway is reported on the left of the graph, the x-axis represents the gene ratio, dot size represents the number of differentially expressed genes present in the pathway and the color indicates the adjusted p.value of the pathway enrichment.”

In figure 6F: “The dotplot shows the significant Gene Ontology Biological processes that emerged from the enrichment analysis of “Reverted DEGs” performed with ClusterProfiler. The name of the pathway is reported on the left of the graph, the x-axis represents the gene ratio, dot size represents the number of differentially expressed genes present in the pathway and the colour indicates the adjusted p.value of the pathway enrichment.”

7. Please provide more information for the CM-H2DCFDA, especially types of ROS which can be detected.

CM-H2DCFDA (5-(and-6)-chloromethyl-2',7'-dichlorodihydrofluorescein diacetate) is a cell-permeable fluorescent probe used to measure reactive oxygen species (ROS) in living cells. The CM-H2DCFDA probe detects reactive oxygen species (ROS), specifically hydrogen peroxide (H₂O₂), as well as other peroxides and hydroxyl radicals (•OH). The probe is commonly used to measure intracellular oxidative stress. Once inside the cell, CM-H2DCFDA is deacetylated to form the non-fluorescent H2DCF, which can then be oxidized by ROS to produce a fluorescent compound (DCF), allowing for detection.

While it detects a range of ROS, it is particularly sensitive to hydrogen peroxide and peroxy radicals, with some reactivity also to hydroxyl radicals and other reactive molecules.

8. Results page 6, lines 28-29: "These results demonstrate that DDR and more specifically tDDR activation, is caused by ROS generated upon exposure of mouse primary neurons to Aβ42 oligomers." In Fig. 3D, E, only 10% of DDR is at telomeres, and more than 50% of DDR was suppressed by NAC, so ROS cause general DDR not specifically tDDR.

Following the referee's comment we edited as follows: "These results demonstrate that DDR including tDDR activation, is caused by ROS generated upon exposure of mouse primary neurons to Aβ42 oligomers." (pag6, lines 36-37)

9. Fig. 5B: Please show % of DDR-positive telomers, and also ROS level should be examined to show that tASOs do not alter ROS production.

These data were removed from the point-by-point response as confidential, but they were provided by the authors during peer review for the information of the editorial team and the reviewers.

Referee #3:

EMBOJ:

Authors: Sepe et al.

Title: Telomeric DNA damage response activation mediates Aβ42 oligomers neurotoxicity in Alzheimer disease

The manuscript presents data showing that in the cortex and hippocampus of 3xTg-AD mice there is an activation of the DNA damage response, specifically at telomeres. The work expands these observations to include iPSC-derived human cortical neurons. Exposure to Aβ42 enhances this response, purportedly due to increased levels of reactive oxygen species. The authors further maintain that this damage is caused by an imbalance in calcium homeostasis. Then, using antisense oligonucleotides against RNA generated at telomeres reduced neuronal death. The authors then proposed that the data point to the potential of RNA-based therapies for Alzheimer's disease.

The Introductory section captures many of the relevant topics and references, but as written it is choppy and would benefit from some editing to smooth the flow of logic. One problem the authors face is that while many of the phenomena described are correlated with one another, the arrows of causality have not been rigorously shown. This is most evident in the linkages between Aβ, ROS, NMDA-R, Ca²⁺, and DNA

damage. Attention to this detail would improve readability

We thank the referee for the suggestion and we modified the text accordingly to the referee's comments in order to make the text more readable and clearer.

To support and better explain the causality claims, we summarize the experiments and results obtained supporting the causal flow of events as summarized in the model in Fig 8:

- following calcium deprivation by EGTA or NMDA/AMPA channel blocker, we observe less intracellular calcium, less ROS and less DDR/tDDR
- following ROS reduction by NAC administration, we observe less DDR/tDDR and improved viability
- following tASO treatment, we observed less DDR/tDDR and improved viability and reduced toxicity of A β 42 oligomers based on RNA seq study.

Figure 1 is described as being from "brain cortex". This is not adequate given that DNA damage is likely variable from one region to the next and at different cortical depths.

We have now included a specification of the brain area of the mouse analysed (pag 4 lines 9-10 and figure legends Fig. 1A-B). They are the cerebral cortex, specifically the isocortex, and the hippocampal formation. The distribution across cortical layers was relatively homogeneous.

The 53BP1 staining image (Fig. 1A) would be better if red and green were used to show overlap of nuclear foci instead of yellow and cyan (also the meaning of the arrows in the figure are not explained in the legend).

Throughout the paper, we have used colour-blind-friendly combinations to ensure accessibility. Specifically, we chose cyan and yellow for this purpose. However, for this referee we show here below images with red and green markers.

Also, we apologize for the incorrect indication of the arrows in the figure. We have corrected the arrows to accurately point to the correct foci signals in the revised images.

The use of γ -H2AX as a DNA damage marker is appropriate but lacks precision. The phosphorylation of H2AX spreads from the break as repair proceeds and with time is spatially less accurate. One possibility to consider is that the modified histone persists at the telomere even after the damage itself has been repaired. A direct assay of a DNA break such as the TUNEL reaction would be an important validation of the claim that damage persists at the telomere.

We acknowledge that there are scientific reports suggesting that γ H2AX may be more a "scar" of past DNA damage events rather than actual frank DNA lesions (Rodier et al, J Cell Sci, 2011). Since this is a relevant concern, in the past we previously explored this possibility and

ruled out by demonstrating that, even in deep senescent cells, DDR foci labelled with antibodies against γ H2AX are associated with unrepaired DSB containing free DNA ends. We demonstrated this *in situ* at individual foci (Galbiati et al, Aging Cell, 2017).

Importantly, this conclusion of ours is supported by an additional independent report in which the very same mouse model that we adopted in this manuscript was tested for the presence of DSB in brain hippocampus by a comet assay. This analysis revealed that 3xTag mice have higher levels of DSB than wt control animals (Sycora et al, NAR, 2015).

The use of p21 as a damage marker is inadequate by itself as it is also a marker of senescence and direct cell cycle activity inhibition.

The p21Waf1/Cip1 protein, encoded by the CDKN1A gene, is primarily regulated by p53 in response to DNA damage and cellular stress. DNA damage activates p53, which binds to the CDKN1A promoter and enhances transcription, leading to cell-cycle arrest. Here we indeed analysed p21 as a marker of downstream activation of DNA damage response in neurons. Thus, p21, when used alongside the other markers employed in the manuscript, serves as a reliable indicator of DNA damage response activation. The referee is right in saying that p21 is also a marker of senescence. But there is no contradiction since we, like many others in the field, believe that DNA damage triggers a DNA damage response that may also lead to cellular senescence. Hence, the factors involved (including p21) are common and can be functionally shared among DNA damage, cellular senescence and cell-cycle inhibition events.

qPCR of message levels loses all spatial and cell-type information. This is important because p21 is elevated in neurons in AD brains with no signs of punctate staining as would be expected if telomeres were labelled (see references from Thomas Arendt).

The referee is correct that spatial information is lost. However, since p21 is regulated at the transcriptional level, measuring its transcript levels provides a precise and quantitative assessment of its engagement. Additionally, since p21 is not actively recruited to sites of DNA damage, it does not form foci, further supporting the use of transcript measurements for its accurate detection. Noteworthy, the dissection of the brain allowed us to select the relevant regions which are mostly composed by neurons.

Regarding Thomas Arendt's research mentioned by the referee, we thoroughly searched his studies and relevant seem:

- *Induction of p21ras in Alzheimer pathology, Neuroreport, 1995*
- *Elevated expression of p21ras is an early event in Alzheimer's disease and precedes neurofibrillary degeneration, Neuroscience, 1999*
- *Linking cell-cycle dysfunction in Alzheimer's disease to a failure of synaptic plasticity. Biochimica et Biophysica Acta, 2007.*

However, Thomas Arendt's papers primarily focuses on p21ras, which is a member of the RAS family of small GTPases, rather than p21Waf1/Cip1, the CDKN1A gene product, which we studied in our paper. p21ras is involved in intracellular signalling pathways that regulate neuronal function, including the MAPK/ERK pathway, which influences synaptic plasticity and survival.

To avoid any confusion, when p21 is first mentioned in the results section of our manuscript, we point the reader to p21 the CDK inhibitor: the product of the CDKN1A gene (page 3 lines 6-7).

Efforts should be made to assay other more direct indices downstream of the DDR - components of the MRN complex for double strand breaks, PARP1 and others for single strand breaks.

MRN complex, which function as early sensors of DNA damage and act upstream in the DNA damage response, unfortunately does not form easily detectable foci at damage sites, thus making their study hard to perform, and virtually impossible in tissues.

In regard to double-strand break markers, 53BP1 and γ H2AX are robust and well-established markers for double-strand breaks. These have been extensively validated in the literature for their specificity in labelling DNA double-strand breaks, including validation by "DNA damage in situ ligation followed by proximity ligation assay" (DI-PLA) which we mentioned earlier, and their use in our study is consistent with standard practices. In addition, an independent study, demonstrated an increase of DNA strand breaks in 3XTgAD brain using the neutral comet assay which identifies double-strand breaks, and this correlates with an increase of γ H2AX and 53BP (Sycora et al, 2015 NAR).

In regard to the detection of single-strand breaks and PARP1 activation, stimulated by this referee, we performed immunocytochemistry using an antibody against poly-ADP-ribose to detect PARylation, the product of PARP activation upon single-strand breaks recognition. The antibody used and staining protocol employed were recommended by the Caldecott Lab, a leading group in the field (Hock et al., Nature, 2017). Under these settings, we did not observe significant differences between wild-type and 3XTgAD mice in either hippocampal and isocortex areas studied – see below.

Additionally, we performed immunocytochemistry staining of the same regions with an antibody against pRPA (ser48), a recognized marker of single-strand breaks that we successfully employed in tissues in a previous report (Gioia et al, Nat Cell Biol, 2022). Again, we did not observe any differences between wild-type and 3xTgAD mice.

Our conclusions are consistent with recent work (Polyzos et al, Nat Comm, 2024) proposing that in the brain, even though oxidative stress may be a major source of DNA damage, single-strand breaks are rapidly converted into double-strand breaks. Overall, all this supports the notion that the type of damage accumulating and being detected and reported in our manuscript is most likely the signal of double-strand breaks.

The Bcl2/BAX ratio of qPCR signals is a very imprecise way of addressing the question of cell death. Even with the dissection to enrich for granule cells there will be many other cell types present, and the source of the signal could be coming from anywhere. Immunocytochemistry would be preferable in this situation.

As requested by the referee, we performed immunofluorescence analysis using antibodies against BAX and BCL-2 proteins. Once again, we observed an increase in the BAX/BCL-2 protein ratio. We have included these analyses in both brain regions, as shown in Fig. 1H, and in the EV Fig.1E (described at page 4 lines 39-42). To better identify neurons in these areas and highlight their distribution, as suggested by this referee, we used MAP2 as a neuronal marker.

Consistently with the analysis of mRNA levels, also the ratio between the signals derived from the stainings for the protein level is increased in the brain of 3xTgAD

The nature of the A β preparation is worrisome. The western blots in EVFig. 2 show no obvious difference between the 6E10 and A11 stain. The presence of a monomeric band in the A11 lane is especially troubling as the antibody purportedly does not label the non-aggregated forms of the peptide.

We agree with the referee that the A11 antibody, which should specifically recognize the oligomeric form, also detects the monomeric band. We hypothesized that this is due to some nonspecific recognition of monomeric forms present in the solution. Upon reviewing the original paper describing this antibody, we noticed that also the authors observed faint monomeric signals when using the A11 oligomeric antibody (Kayed et al, 2010 Molecular Neurodegeneration), and this depended on the amounts of oligomers loaded on the gel. To further investigate this ourselves, we performed a western blot using both the stock concentration (100 μ M) of our A β 42 oligomer solution, as shown in the submitted version of the manuscript, and the final concentration reached in the neuronal culture medium (10 μ M). We observed that at the lower concentration tested, the monomeric forms were no longer detectable with the A11 oligomeric antibody and was even less detectable with the A β E10 antibody, as shown in the images below. This suggests that, not uncommonly, the nonspecific

binding of the antibodies increases with the concentration of A β 42 oligomers. For this reason, in the revised manuscript, we have replaced the original Western blot images with the new one in which the solution was loaded at a final concentration of 10 μ M, as used in the experiment (EV Fig. 2C). It is evident that in this case, the pattern of the two antibodies is different: while A β E10 detects all bands, the A11 oligomeric antibody primarily recognizes the oligomeric forms.

We thank the referee for spotting this.

Further, the application of A β increases γ -H2AX staining, but close inspection of Fig. 2B reveals that the location is adjacent to, not overlapping with, the telomeric labelling. This tends to undercut the authors argument and should be clarified. Again, the meaning of the white arrows is not explained and is not obvious from their location.

In response to the referee's comments, we have replaced Fig.2B with a better image that more clearly shows the colocalization events between γ H2AX and telomeric signals.

We want to emphasize that the images shown are representative and result from the flattening of multiple focal planes (z-stack) acquired using a confocal microscope, but they were not used for the identification of the colocalization events. The colocalization analysis was performed analysing individual focal planes to detect colocalization events. For this we used a Fiji/ImageJ pipeline specifically designed by our imaging facility for 3D imaging. This pipeline analyses each focal plane independently applying a threshold of at least 10-20 contiguous voxels to detect a colocalization event.

We believe these clarifications, along with the updated images, adequately address the referee's concerns.

The RNAseq data is confusing. The X-axis of the volcano plot in Fig. 2G is compressed, presumably to include the four outlier points, but the impression is that with the exception of these four, there is very little change in gene expression. The high significance is noted, but the finding that gene expression is little changed is surprising given that application of oligomers is such a strong stimulant. The genes that are most changed or most significant should be identified.

Following the referee request, we have modified the Volcan Plots to display a clearer representation of the gene expression changes (Fig 2G and Fig 6D).

In regard to the reported gene expression changes, in the manuscript submitted we have described that there are 549 significantly differentially expressed genes (page 6 line 2). Now, stimulated by the referee comment, we added in the revised manuscript an additional Gene

Ontology biological process analysis showing the several pathways impacted by A β 42 oligomers (EVFig.4A described at page 6 lines 1-4).

Consulting the Agora data makes sense, but the brain regions identified are not revealed, nor are they validated in the mouse brain. This diminishes the power of the analysis.

Fig. EV6B is also a confusing presentation of the data. The final two columns are consistent with the authors' claims about the effect of the ASO, but controls for the effects of ASO alone are missing (no A β). The rationale for showing the brain region data is not clear as there is no clear relationship between the two right hand columns and the rest.

We apologize for the lack of clarity. The full names of the brain regions identified were provided in the figure legends and methods.

The referee is correct in noting that a direct comparison between mouse brain and Agora datasets has not yet been performed. For this reason, we decided to apply a stringent threshold considering “concordant DEGs” the DEGs emerged from our RNA-seq that change consistently in more than half of the regions identified in the Agora database. We have clarified the parameters used to obtain these results in the text (in the result section at page 6 lines 15-17 and M&M at page 18 lines 8-11).

Part of these data were removed from the point-by-point response as confidential, but they were provided by the authors during peer review for the information of the editorial team and the reviewers.

Overall, a considerable amount of work has gone into this study, but the presentation and the analysis are flawed in important ways and the strength of the support for the authors' hypotheses is therefore diminished.

We believe that our explanations, arguments and additional analyses and results have addressed the referee's concerns effectively and significantly strengthened the manuscript, further support our hypothesis.

Minor points:

When discussing replicates, the authors should clarify whether these are technical or biological replicates (multiple coverslips from one culture or multiple, independent, cultures)

Thank you for this comment. The replicates mentioned in our manuscript are biological replicates. Specifically, biological replicates refer to samples obtained from independent biological experiments, meaning that, for example, RNA samples and coverslips, used for RT-qPCR and immunofluorescence analysis, derived from different cultures of mouse primary, each originating from distinct pools of pups. Additionally, different solutions of A β 42 oligomers or ASO were used in these independent experiments. We add in the statistical paragraph the explanation of biological replicates (page 19 lines 5-8)

The abstract mentions human iPSC neurons, but nearly all of the work is done with mouse cells. A restructuring of the text would be advised.

We have modified the text accordingly to the referee request.

The involvement of cell death should be monitored in more experiments. The DAPI images (e.g., Fig. 4C suggest that the oligomers induced a fair amount of apoptosis. If correct, this would alter the interpretation of the findings.

The DAPI-positive debris in the images are a common occurrence in culture of mouse primary neurons, indeed they are observed to similar extents in all conditions and they do not correlate with changes in cell viability as measured in our viability assays and Bax/Bcl-2 levels. For comparisons we show below images from other studies which show the same pattern indicated by the white arrows that we add for the referee.

Suberbielle et al, Nature Neuroscience, 2013

Cheng et al, Scientific Report, 2015

Dear Fabrizio,

Thank you again for submitting your revised manuscript (EMBOJ-2024-119642R) to The EMBO Journal for our consideration, and for your patience during peer review. As I have already informed you, your manuscript was sent back to the original referees who had previously assessed the initial version of your work, and we have received their comments, which I have already shared with you (they are included again below). I would also like to thank you for your draft point-by-point response to the remaining comments of referee #3, which was very helpful for us to reach a fair and balanced decision on your manuscript.

Referees #1 and #2 recognize that the manuscript has been sufficiently revised and their initially raised concerns have been satisfactorily addressed, and they now recommend publication of the manuscript without any further comments. Referee #3, on the other hand, finds the revision commendable but still raises a number of remaining criticisms and concerns.

On balance, and in light of both this expert input and your draft point-by-point response to the remaining comments of referee #3, I would like to invite you to submit a final version of your manuscript taking on board and addressing the comments of referee #3, along the lines of your draft point-by-point response. Please submit along with your revised manuscript a detailed point-by-point response to the remaining comments, which will be published online (included in the Peer Review File) along with your article. I would like to mention that we do agree with referee #3 that the Introduction would benefit from more clarity and a better flow of the text, which we kindly invite you to revise accordingly.

From the editorial side, there are also a few corrections and changes we need you to make in the final version of the manuscript before we can proceed with its acceptance for publication in The EMBO Journal:

- Please make sure that the profile in our manuscript tracking system of the co-author Alessia di Lillo is updated so that the first and last names are properly entered in the profile.
- The funding information included in the Comments box could not be extracted by our production team, and therefore all funders should be added to the "More Funders" list; the Funding information in the manuscript file should be included in the Acknowledgements section.
- Please provide a list of up to 5 relevant keywords after the Abstract of your revised manuscript to improve search engine discoverability of the article.
- The reference format in your manuscript is not consistent with our journal's style; the list should be alphabetical, with the names of only the first 10 co-authors of each citation listed and followed by "et al.". Please see our guide to authors for more information: <https://www.embopress.org/page/journal/14602075/authorguide#referencesformat>.
- Please make sure that the full and permanent access information for your deposited RNA sequencing data is provided in the Data availability statement, and also make sure that the data will be publicly available at the time of publication.
- The "Competing interests" heading should be renamed to "Disclosure and competing interests statement".
- The author contributions statement should be removed from the manuscript file. Instead, we use CRediT to specify the contributions of each author in the journal submission system. Please feel free to use the free text box to provide more detailed descriptions during submission. See also our guide to authors for more information: <https://www.embopress.org/page/journal/14602075/authorguide#authorshipguidelines>.
- Source file names, titles, legends and manuscript callouts all need to be updated to "Figure EV1-EV6" instead of "Expanded View Fig. 1-6".
- All Figure callouts should be listed sequentially.
- We noticed that callouts for Expanded View (EV) Figures are missing (there is only EV Fig. 2A).
- There are several mistakes in the Author Checklist that should be corrected; in particular, there are two "Yes" responses in the "Core facilities" section for only one question; in the "Experimental animals" section, the manuscript section where the information can be found must be indicated in the last column of the sheet; and there are several typos throughout the checklist.
- Please upload the main and Expanded View (EV) Figures as individual high-resolution Figure files. Please see our guide to authors for more information: <https://www.embopress.org/page/journal/14602075/authorguide#figureformat>.
- Materials and methods need to be described in the manuscript using our structured methods format, which is now required for all research articles. According to this format, the Methods section includes a single "Reagents and Tools Table" -listing key

reagents, experimental models, software and relevant equipment including their sources and relevant identifiers- followed by a "Methods and Protocols" section describing the methods. Please download and fill our Reagents and Tools Table template (.docx), which you can find in our author guide: <https://www.embopress.org/page/journal/14602075/authorguide#structuredmethods>. When submitting your revised manuscript, please do not include the Reagents and Tools Table in the Methods section of the manuscript but instead upload it as a separate file choosing the file type "Reagent Table".

- Please note that EMBO press papers are accompanied online by:

A) a short (2 sentences) summary of the findings and their significance,

B) 2-5 short bullet points highlighting the key results, and

C) a synopsis image in .jpg or .png format that is exactly 550 pixels wide and 300-600 pixels high (the height is variable). Please note that the text needs to be legible at the final size.

Please upload this information along with your revised manuscript (the text for A and B should be provided in a separate Word file).

- During our standard Figure checks, we detected signs of potentially suboptimal processing of the Western blots shown in Figure EV2B that we would need to investigate further. Please include in your Source Data the original, uncropped and unedited blots (of all available replicates) for these experiments.

- During our routine data checks, our data editors have raised the following queries regarding figures, data, and legends. Please make sure that all requests below are completely addressed in the final version of your manuscript:

1. Please note that the exact p values should be provided in the legends of Figures 3F, 7F, G (or in the Figures themselves).

2. Please indicate the statistical test used for data analysis in the legends of Figures 2G, I; 6D, F; EV4 A, C; EV6.

3. Please note that information related to "n" is missing in the legends of Figures 2G, 6D.

4. Please note that the scale bar needs to be defined in the legends of Figures 3A, 4A, B; EV5 A.

5. Please note that scale bar and its definition in the legends are missing for Figures EV3 A, C; EV2 C.

6. Please note that the white arrows are not defined in the legend of Figure 1A. This needs to be rectified.

- The order of the manuscript sections must be corrected as follows: Title page - Abstract and Keywords - Introduction - Results

- Discussion - Methods - Data Availability - Acknowledgements - Disclosure and Competing Interests Statement - References -

Figure Legends - main Tables (if there are any) - Expanded View Figure Legends.

Please also note that as part of the EMBO publications' Transparent Editorial Process, The EMBO Journal publishes online a Peer Review File along with each accepted manuscript. This File will be published in conjunction with your paper and will include the referee reports, your point-by-point response and all pertinent correspondence relating to the manuscript. You can opt out of this by letting the editorial office know (contact@embojournal.org). If you do opt out, the Peer Review File link will point to the following statement: "No Peer Review File is available with this article, as the authors have chosen not to make the review process public in this case."

We look forward to seeing a final version of your manuscript as soon as possible. Please let us know if you have any questions and use this link to submit your revision: xxxxxxxxxxxx.

Best regards,

Ioannis

Referee #1:

The reviewers have satisfactorily replied to my concerns.

Referee #2:

The authors appropriately responded to the referee's concerns as well as to the other referees' concerns. Now this manuscript is suitable to be published in EMBO Journal.

Referee #3:

EMBOJ-2024-119642R

Authors: Sepe et al.

Title: Telomeric DNA damage response activation mediates A β 42 oligomers neurotoxicity in Alzheimer disease

The authors are to be commended for their efforts to revise the work. The response letter is detailed and represents a good faith attempt to address the criticisms of the three reviewers. Unfortunately, the work is still deficient in many regards and the conclusions put forward do not seem justified by the data.

The Introductory section is improved but remains choppy as it jumps from topic to topic. The model and the included diagram are helpful, but the flow of logic is still not as clear as it could be

The use of yellow and cyan is well justified (although this reviewer appreciated the red/green figure in the rebuttal). However, the colocalization of telomeric markers and DNA damage markers is unconvincing. The past publications of the group are acknowledged, but it is unclear how the yellow and magenta dots in Figure 1B can be viewed as co-localized or not. There will certainly be variability in any staining method, but the density of "telomeres" and "centromeres" should at least be similar from one preparation to the next. That does not apply to the images presented. The arrows in the 3xTG panel are still offset (the top one points to nothing) and the most convincing co-localizations are found in the wt image. Since one would assume that these images are the best examples of the material, the quantification presented in the later panels rests on an uncertain foundation. It is not sufficiently justified to assert as the authors do on line 23 that "the DDR markers preferentially accumulate at telomeres."

The use of p21 as a damage marker remains inadequately justified although the point is well taken that the Arendt references measured a different protein. (He nonetheless used other CKIs and did find similar elevations in AD neurons.)

The Bcl2/BAX ratio of qPCR signals remains an imprecise way of addressing the question of cell death. The use of immunocytochemistry is a help, but it is still unclear why a simpler marker such as cleaved caspase-3 was not used.

The authors have responded to the concerns raised about colocalization determination, but doubts remain. Figure 2B now does indeed show "white" dots where telomeric and γ -H2AX staining overlap, but the density of the magenta telomeric marker is so high that some overlap is expected on a purely random basis. The authors claim that their images are flattened Z-stacks of confocal images. After the previous concerns raised it would seem logical to present a single optical plane to make this case more solid. As it stands, the data do not support the claim made in the title of the paper that it is the DNA damage at the telomeres that induces the neurotoxicity.

Referee #3:

EMBOJ-2024-119642R

Authors: Sepe et al.

Title: Telomeric DNA damage response activation mediates A β 42 oligomers neurotoxicity in Alzheimer disease

The authors are to be commended for their efforts to revise the work. The response letter is detailed and represents a good faith attempt to address the criticisms of the three reviewers. Unfortunately, the work is still deficient in many regards and the conclusions put forward do not seem justified by the data.

The Introductory section is improved but remains choppy as it jumps from topic to topic. The model and the included diagram are helpful, but the flow of logic is still not as clear as it could be.

We thank the reviewer for recognizing the improvements made to the introduction and for their valuable feedback on its clarity and flow. We modified in the revised manuscript the text accordingly to the referee's comments to improve the flow.

The use of yellow and cyan is well justified (although this reviewer appreciated the red/green figure in the rebuttal). However, the colocalization of telomeric markers and DNA damage markers is unconvincing. The past publications of the group are acknowledged, but it is unclear how the yellow and magenta dots in Figure 1B can be viewed as co-localized or not. There will certainly be variability in any staining method, but the density of "telomeres" and "centromeres" should at least be similar from one preparation to the next. That does not apply to the images presented.

The arrows in the 3xTG panel are still offset (the top one points to nothing) and the most convincing co-localizations are found in the wt image. Since one would assume that these images are the best examples of the material, the quantification presented in the later panels rests on an uncertain foundation. It is not sufficiently justified to assert as the authors do on line 23 that "the DDR markers preferentially accumulate at telomeres."

We acknowledge the reviewer's concern regarding the interpretation of yellow and magenta overlap in Figure 1B. To improve clarity, here below we now show the same images using red and green channels with additional arrows pointing to colocalization events. In the revised manuscript, we have included an explicit labeling of the nuclei, as we did for the referee in the image below, to enhance the clarity of the figure. Additionally, we have correctly positioned the arrows to indicate the colocalization events also shown in the red and green channels images included below for the referee. We apologize for the incorrect placement of the arrows in Figure 1B in the previous version (which occurred during the export to PDF). Dashed circles represent nuclei.

Importantly, these pictures aside, quantifications in Figures 1C and 1D were performed computationally and on a large number of nuclei (> 50) from each animal for a total of 200-500 cells. Colocalization analyses were performed on individual focal planes by using an in-house generated and validated Fiji/ImageJ pipeline specifically designed in collaboration with our imaging facility. This pipeline analyzes automatically each focal plane independently applying a threshold of at least 10-20 contiguous voxels to detect a colocalization event. Thus, quantifications are objective, accurate and unbiased. We would also clarify that images shown were the most representative, rather than the “best looking”.

Finally, the number of telomeres per nucleus and centromeres analyzed in each group was consistent across samples. The graph below illustrates the distribution of ROIs (regions of interest) of telomeres and centromeres, and it demonstrates similar density of telomeres and centromeres per nucleus – just like expected by this referee.

In light of the above clarifications and additional supporting data, we maintain that our statement on line 23 "DDR markers preferentially accumulate at telomeres" is justified. Notably, this conclusion is independently supported by the detection of tncRNA.

The use of p21 as a damage marker remains inadequately justified although the point is well taken that the Arendt references measured a different protein. (He nonetheless used other CKIs and did find similar elevations in AD neurons.)

p21 (CDKN1A) induction is widely recognized as a result of DNA damage generation and it is part of the DNA damage response (DDR) (Jurk et al, Ageing cell, 2012, Hou et al, Nat Rev Neurol, 2019). Its expression is primarily regulated by p53, which is activated upon DNA damage by the DDR kinases ATM and ATR. Thus, increased p21 levels is confidently used to indicate DDR activation. Noteworthy, p21 study was shown here in combination with other robust DNA damage markers such as γ H2AX and 53BP1 that, altogether makes us confident that we are characterizing DNA damage-related events. Finally, we agree with this referee that also other publications, including those by Thomas Arendt, report the induction of CDK inhibitors – of which p21 is one of them.

The Bcl2/BAX ratio of qPCR signals remains an imprecise way of addressing the question of cell death. The use of immunocytochemistry is a help, but it is still unclear why a simpler marker such as cleaved caspase-3 was not used.

As previously noted by the referee, we have addressed this point by presenting the cell distribution in brain sections of the BAX and BCL2 proteins to support our RNA results. We would like to stress that the primary objective of our study is not to demonstrate neuronal death in these mice, since this remains a contentious issue, especially at this age – consider that for instance that Sykora et al (NAR, 2015) report some evidence of cell death which however does not reach statistical significance till 24 months of age (our mice are 12 months of age).

Additionally, it is important to consider that cleaved caspase-3, although commonly used as a marker of apoptosis, in the context of early stages of Alzheimer's disease has been shown to also perform non-apoptotic functions (as discussed in studies by D'Amelio et al, Cell Death and Diff, 2010). For this reason, we chose not to employ this marker.

To reiterate: our study's focus is to emphasize that DNA damage correlates with early neuronal dysfunction, including the induction of pro-apoptotic genes, in a context where widespread neuronal death is not yet evident.

The authors have responded to the concerns raised about colocalization determination, but doubts remain. Figure 2B now does indeed show "white" dots where telomeric and γ -H2AX staining overlap, but the density of the magenta telomeric marker is so high that some overlap is expected on a purely random basis. The authors claim that their images are flattened Z-stacks of confocal images. After the previous concerns raised it would seem logical to present a single optical plane to make this case more solid. As it stands, the data do not support the claim made in the title of the paper that it is the DNA damage at the telomeres that induces the neurotoxicity.

In response to this referee's comment, we show here below a representative single optical plane image of three individual neurons treated with A β oligomers. The white arrows (top panel) indicate colocalization events. The lower panel shows the same neurons stained with DAPI (DNA) and MAP2 (a neuronal marker). Notably, the observed overlap is among distinct, sharp, well-defined focal signals, without background signal and nonspecific staining.

We would also like to emphasize that our conclusions are not solely based on the images presented in the manuscript, but they are supported by comprehensive statistical analyses of over 50 nuclei per replicate, examined across at least three independent biological replicates for a total of 150-200 cells. These analyses demonstrate a significant increase in telomeric DNA damage in A β treated neurons compared to vehicle-treated ones or other genomic regions with similar characteristics (centromeres). Importantly, all quantifications were performed using automated software (as described earlier), eliminating potential human bias. The images presented are those most representative of the observed effect, not selected for visual appeal.

Furthermore, the colocalization analysis was independently supported by the increase of tncRNA.

To further substantiate the role of telomeric DNA damage in A β oligomer-induced neurotoxicity, we provide robust data showing that specific inhibition of the DNA damage response (DDR) at telomeres, by tASO, leads to a notable reduction in DDR markers (53BP1, p21), improved neuronal viability, and a shift towards anti-apoptotic gene expression. Additionally, RNAseq analysis shows that the tASO reverses A β -induced pathway changes, many of which are also associated with Alzheimer's disease in patients.

Dear Fabrizio,

Congratulations on an excellent study! I am very pleased to inform you that your manuscript has been accepted for publication in The EMBO Journal. Thank you for comprehensively addressing the referees' initially raised criticisms and concerns, and the editorial requests for changes and corrections.

If you have any questions, please do not hesitate to contact the Editorial Office. Thank you for your contribution to The EMBO Journal. Working with you has been a pleasure!

Best regards,

Ioannis

** Click here to be directed to your login page: xxxxxxxxxxxxxxxxxxxxx